

**Elucidating real-world vehicle emission factors from mobile**
**measurements over a large metropolitan region: a focus on isocyanic**
**acid, hydrogen cyanide, and black carbon**
Sumi N. Wren, John Liggio, Yuemei Han, Katherine Hayden, Gang Lu, Cris M. Mihele, Richard L.
Mittermeier, Craig Stroud, Jeremy J. B. Wentzell, Jeffrey R. Brook
Air Quality Process Research Section, Air Quality Research Division, Environment and Climate
Change Canada, 4905 Dufferin St., Toronto, ON, M3H 5T4
Corresponding Authors: Sumi N. Wren (sumi.wren@gmail.com), Jeffrey R. Brook
(jeff.brook@canada.ca)



## 1 Abstract

A mobile laboratory equipped with state-of-the-art gaseous and particulate instrumentation
was deployed across the Greater Toronto Area during two seasons. A high-resolution time-of-flight
mass spectrometer (HR-TOF-CIMS) measured isocyanic acid (HNCO) and hydrogen cyanide (HCN),
and a high-sensitivity laser-induced incandescence (HS-LII) instrument measured black carbon
(BC). Results indicate that on-road vehicles are a clear source of HNCO and HCN, and that their
impact is more pronounced in the winter, when influences from biomass burning and secondary
photochemistry are weakest. Plume-based and time-based algorithms were developed to calculate
fleet-average vehicle emission factors (EF); the algorithms were found to yield comparable results,
depending on the pollutant identity. With respect to literature EFs for benzene, toluene, C2 benzene
(sum of m,p,o-xylenes and ethylbenzene), nitrogen oxides, particle number concentration (PN), and
black carbon, the calculated EFs were characteristic of a relatively clean vehicle fleet dominated by
light-duty vehicles. Our fleet-average EF for BC (median: 25 mg $kg_{fuel}^{-1}$, interquartile range: 10 – 76
mg $kg_{fuel}^{-1}$) suggests that overall vehicular emissions of BC have decreased over time. However, the
distribution of EFs indicates that a small proportion of high-emitters continue to contribute
disproportionately to total BC emissions. We report the first fleet-average EF for HNCO (median:
2.3 mg $kg_{fuel}^{-1}$, interquartile range: 1.4 – 4.2 mg $kg_{fuel}^{-1}$) and HCN (median: 0.52 mg $kg_{fuel}^{-1}$,
interquartile range: 0.32 – 0.88 mg $kg_{fuel}^{-1}$). The distribution of the estimated EFs provides insight
into the 'real-world' variability of HNCO and HCN emissions, and constrains the wide range of
literature EFs obtained from prior dynamometer studies. Our results demonstrate that although
biomass burning is a dominant source of both air toxics on a national scale, vehicular emissions
play an increasingly important role at a local scale, especially in heavily-trafficked urban areas. The
impact of vehicle emissions on urban HNCO levels can be expected to be further enhanced if
secondary HNCO formation from vehicle exhaust is considered.

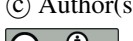



## 1. Introduction

In urban areas, traffic-related air pollution (TRAP) is associated with adverse impacts on human health, air quality, climate change, and the environment (Pope and Dockery, 2006;Grahame et al., 2014;HEI Panel, 2010). Studies of TRAP, from both the emission and exposure perspective, have often focused on criteria air pollutants such as nitrogen oxides ($NO_x$), carbon monoxide (CO) and particulate matter (PM) (Jerrett et al., 2009;Beckerman et al., 2008). However, it is not established if these species are directly responsible for negative outcomes associated with TRAP, or if they act in tandem with, or as proxies for, other compounds in the pollutant mixture (Brook et al., 2007;Mauderly and Samet, 2009;Dominici et al., 2010). For that reason, it is imperative that other components of TRAP are characterized, including the near-road exposures and vehicle emission factors of additional air toxics. In the current study we focus on vehicle emissions of black carbon (BC), isocyanic acid (HNCO) and hydrogen cyanide (HCN).

Although particulate mass is often used as an indicator for health risks associated with combustion, it has been suggested that black carbon may be a more effective metric (Janssen et al., 2011;Grahame et al., 2014). Black carbon particles pose a significant health risk due to their chemical stability, large surface area, and small mode diameter, with the smallest (i.e., 'ultrafine') BC particles able to penetrate the lung lining and enter the blood stream (Highwood and Kinnersley, 2006). However, it is not established whether it is the compounds associated with BC (such as particle bound polycyclic aromatic hydrocarbons) or the BC itself that are responsible for negative effects (Janssen et al., 2011). The dominant sources of BC are combustion-related and include open biomass burning (BB) and residential, industrial and transportation-related fossil-fuel burning. Anthropogenic BC emissions have been closely linked to vehicle emissions, particularly those associated with heavy-duty diesel vehicles (HDDV) (Bahadur et al., 2011;Ban-Weiss et al., 2008). Although, BC emissions from light-duty gasoline vehicles (LDGV) have been considered to be quite low by comparison, their exact magnitude is not well constrained, with recent studies suggesting that they may both be underestimated (Liggio et al., 2012;Krecl et al., 2017) and overestimated (Wang et al., 2016). Furthermore, improvements in emissions control technologies have seen HDDV BC emissions decrease significantly (Dallmann et al., 2012;Krecl et al., 2017). As a result, the relative importance of gasoline versus diesel engines as sources of BC is not well established, leading to uncertainties in present-day on-road inventories (Liggio et al., 2012;Krecl et al., 2017). Given the rapid pace of change of fuel injection and emission control technologies, establishing current, fleet-average BC emission factors (EF) is important for evaluating bottom-up inventories, which are necessary from both a health/air quality and climate perspective (Bond et al., 2013).

Only recently has it been suggested that HNCO (Wentzell et al., 2013;Brady et al., 2014;Link et al., 2016;Suarez-Bertoa and Astorga, 2016;Jathar et al., 2017) and HCN (Crounse et al., 2009;Moussa et al., 2016;Harvey et al., 1983) can be emitted by on- and off-road vehicles. Isocyanic acid is a highly toxic gaseous acid which dissociates at physiological pH to form cyanate anions ($NCO^-$) which in turn participate in damaging carbamylation reactions, thereby leading to adverse health outcomes such as cataracts, atherosclerosis, rheumatoid arthritis, cardiovascular disease, and renal failure (Roberts et al., 2011 and references therein). Roberts et al. (2011) used the



physical properties of HNCO to estimate that ambient mixing ratios as low as 1 ppbv could be harmful to humans. Similar to HNCO, hydrogen cyanide is a highly toxic gas with known negative effects on human health due to its interference in aerobic metabolism (Logue et al., 2010;Barillo, 2009; U.S. EPA, 2010).

Historically, biomass burning was considered to be the dominant global source of both HNCO (Veres et al., 2010;Roberts et al., 2011;Young et al., 2012) and HCN (Li et al., 2000;Li et al., 2003;Li et al., 2009;Shim et al., 2007). Global HCN (Li et al., 2003) and HNCO (Young et al., 2012) models have hitherto considered vehicle sources of these compounds to be negligible. As such, past measurements of these species have focussed on regions heavily influenced by biomass burning, or in the case of HCN, on the upper troposphere or total tropospheric column. Although the advent of chemical ionization mass spectrometers has allowed for real-time measurements of atmospherically relevant concentrations of these species (Roberts et al., 2011;Veres et al., 2008;Woodward-Massey et al., 2014;Le Breton et al., 2013;Knighton et al., 2009), there remain relatively few measurements of ambient HNCO (Roberts et al., 2011;Roberts et al., 2014;Wentzell et al., 2013;Zhao et al., 2014;Woodward-Massey et al., 2014;Sarkar et al., 2016;Chandra and Sinha, 2016; Kumar et al., 2018). Measurements of ground-level HCN in both rural and urban environments with minimal BB influence are more limited (Ambrose et al., 2010). However, given the recent studies suggesting that HCN and HNCO emissions from vehicles could be significant, especially at a local scale, a better understanding of on-road emissions of these species is necessary. Moreover, ambient measurements are suggestive of a secondary source of HNCO (Roberts et al., 2011;Wentzell et al., 2013;Roberts et al., 2014;Zhao et al., 2014;Sarkar et al., 2016; Kumar et al., 2018) – formed photochemically by the photooxidation of precursors such as alkyl amines and amides (Borduas et al., 2013;Borduas et al., 2015;Sarkar et al., 2016). Recent studies (Jathar et al., 2017;Link et al., 2016) show that diesel engine exhaust itself contains precursors leading to enhanced photochemical production of HNCO, even further underscoring the need to quantify vehicular emissions of HNCO in dense, urban environments.

Existing literature values for HNCO and HCN emission factors have been exclusively obtained from chassis or engine dynamometer studies on a limited number of engines/vehicles. While the strength of dynamometer studies is control over factors such as vehicle age, fuel composition, type of after-treatment technologies, temperature, and driving mode, they have limitations with respect to yielding representative emission factors, for the precise reason that mobile emissions have been shown to be sensitive to such factors (Franco et al., 2013). It is important that the accuracy of emission inventories derived from dynamometer results are verified against in-use vehicle emissions (Parrish, 2006), since emission inventories are often used to constrain regional budgets and exposure estimates for traffic-related air pollutants. This is particularly relevant for HNCO and HCN, where there are large discrepancies in reported emission factors. Although real-world EF measurements can suffer from their own shortcomings (namely lower precision and repeatability), they are essential in identifying gaps and providing insight into actual emission behaviour of on-road vehicles (Franco et al., 2013).

In the present study, we deploy a mobile laboratory over a large metropolitan region in two seasons, with the goal of characterizing near-road exposure and fleet-average emission factors for



black carbon, HNCO, and HCN. These species are discussed alongside benzene, a regulated traffic
pollutant of interest due to its carcinogenic nature, and whose behaviour has been more thoroughly
characterized. The focus in this paper is on the development of plume-based and time-based
methodologies to calculate fuel-based vehicle emission factors.  We assess their performance
against each other and in comparison to available literature EFs for a wide range of pollutants:
benzene, toluene, C2 benzenes (sum of m,p,o-xylenes and ethylbenzene), $NO_x$ (=NO + $NO_2$), particle
number concentration (PN), and black carbon. We report, to our knowledge, the first real-world,
fleet-average HNCO and HCN vehicle emission factors and use them to help assess dynamometer
results relative to real-world conditions. Finally, the estimated fleet-average emission factors are
scaled-up to determine the relative importance of vehicle emissions of HNCO and HCN.
**2.  Materials and Methods**
**2.1. Mobile laboratory measurements – CRUISER**
13       **2.1.1.   Overview of mobile measurements**

14       Air quality and meteorological measurements were made from Environment and Climate
Change Canada's mobile laboratory: Canadian Regional and Urban Investigation System for
Environmental Research (CRUISER) (Levy et al., 2014). CRUISER was deployed during two seasons
over the Greater Toronto Area (GTA), a metropolitan area encompassing the city of Toronto and
four regional municipalities with a population of over 6 million. The Summer Campaign took place
over 9 days in July, 2015 (July 15, 16, 17, 20, 21, 22, 23, 27, 28) as part of the Environment Canada
Pan and Parapan American Science Showcase (ECPASS) (Joe et al., 2018). The Winter Campaign
took place over 8 days in January, 2016 (January 11, 13, 14, 15, 18, 19, 20, 21) as part of a health
exposure mapping study. Driving took place on weekdays only, with the majority of measurements
occurring between 09:00 and 17:00 local time.  Driving routes were chosen to pass along highways,
major roadways, and local streets, and to visit residential, commercial, and industrial areas; the
driving routes for the Summer and Winter Campaign are shown in Supplement Fig. S1. In 2016 the
Ontario vehicle fleet was composed of approx. 97% light-duty (LD) vehicles (vehicles < 4500 kg and
motorcycles/mopeds) and 4% heavy-duty (HD) vehicles (for this paper, the HD category includes
both medium-duty vehicles 4500 – 14 999 kg and heavy-duty trucks > 15 000 kg, and buses)
(Statistics Canada); the composition of the GTA vehicle fleet is assumed to be similar.

30       Several gas phase and particle phase instruments were housed on-board CRUISER as listed
in Table 1. Carbon dioxide ($CO_2$) was measured with 2 s time resolution by cavity-enhanced laser
absorption spectroscopy (PICARRO). All gas phase instruments sampled from a common inlet with
the exception of the high-resolution time-of-flight chemical ionization mass spectrometer (HR-TOF-
CIMS) which sampled off a dedicated inlet located on the roof of CRUISER towards the rear right-
side. The common gas phase inlet was located 3.6 m a.g.l. and oriented near the front left side.
Ambient air was sampled through a 2 m long PFA tube with 0.61 cm ID followed by a 30 cm long
PFA tube with 0.52 cm ID at a rate of 13.6 slpm; inlet lines for various instruments were connected
downstream of this common inlet. All particle phase instruments sampled off a common stainless
steel inlet located adjacent to the gas phase inlet. During the Winter Campaign, $CO_2$ was sampled
from the same inlet as the HR-TOF-CIMS.  Relative wind speed and wind direction was measured
using an ultransonic anemometer located on the roof at the front of CRUISER. Periods of potential



self-sampling were identified and removed using an algorithm which is described in the
Supplement.

### 2.1.2.  Proton-transfer reaction time-of-flight mass spectrometry (PTR-TOF-MS)

Volatile organic compounds (VOC) were measured using a proton transfer reaction time-of-
flight mass spectrometer (PTR-TOF-8000, Ionicon Analytik). The operating principles of the PTR-
TOF-MS instrument have been described elsewhere (Jordan et al., 2009;Li et al., 2017); further
details can be found in the Supplement. The PTR-TOF-MS sampled off the common gas inlet with a
time resolution of 1 s and the response of the PTR-TOF-MS to specific VOCs was determined using a
home-built zero/calibration unit and a custom VOC gas standard (Ionicon). The sensitivities and
detection limits are listed in Table S1.

### 2.1.3.  High resolution time-of-flight chemical ionization mass spectrometer (HR-TOF-CIMS)

HNCO and HCN were measured using a high-resolution time-of-flight chemical ionization
mass spectrometer (HR-TOF-CIMS, Aerodyne Research, Inc.). The design, operation and mobile
deployment of the HR-TOF-CIMS has been previously described (Veres et al., 2008;Roberts et al.,
2011;Wentzell et al., 2013;Liggio et al., 2017a). Additional details can be found in the Supplement.
Briefly, the HR-TOF-CIMS is a differentially pumped time-of-flight mass spectrometer configured to
use iodide ion as the reagent ion, with a time-resolution of 1 s (Woodward-Massey et al., 2014;Le
Breton et al., 2013). Details regarding the calibration, sensitivities, and detection limits can be
found in the Supplement.

### 2.1.4.  High-sensitivity laser-induced incandescence (HS-LII) for Black Carbon

Black carbon measurements were made with a high-sensitivity laser-induced
incandescence (HS-LII) instrument (Atrium Technologies Inc., CA, USA) developed in collaboration
with the National Research Council Canada (NRC). The particular instrument on CRUISER is a
research-grade prototype capable of ultra-low BC measurements at 1 s resolution. Here, black
carbon is operationally defined by its high thermal stability (Petzold et al., 2013). The principle of
operation of this instrument, as well as its use during ambient studies, has been described
elsewhere (Snelling et al., 2005;Chan et al., 2011;Liggio et al., 2012). Briefly, ambient particles
within a set volume are rapidly heated by a pulsed laser beam (1064 nm; 7 ns FWHM, 200
mJ/pulse) to just below the soot sublimation temperature (~4000 K). The absolute incandescence
and temperature of the BC particles are measured by collection optics and photomultipliers. After
appropriate calibration and analysis, these two parameters are used to determine the soot volume
fraction, which is converted to a BC mass concentration with knowledge of the particle material
density ($\rho$) and the absorption function ($E_m$), both of which are well-established for BC (Coderre et
al., 2011;Choi et al., 1994;Wu et al., 1997). Previous studies have shown that the HS-LII can detect
laboratory generated particles < 7 nm in diameter (Stirn et al., 2009). The HS-LII was only in
operation for the Summer Campaign.

### 2.2. Calculating fleet-average emission factors from a mobile platform





Mobile measurements (Jiang et al., 2005;Canagaratna et al., 2004;Zavala et al., 2006;Zavala
et al., 2009;Park et al., 2011;Liggio et al., 2012;Hudda et al., 2013;Jimenez et al., 2000) of individual
tailpipe emissions (i.e. plumes) have proven to be an effective approach for determining fleet
emission factors, with the advantage of covering a large geographical region, while measuring
emissions in real-time over a range of driving modes. Hence they are able to evaluate the
applicability of EF measurements made at a fixed location to the entire region, providing insight
into the degree of emissions variability and identifying the presence of high-emitting vehicles.
Plume-based emissions measurements can be made in two ways: a targeted approach in
which individual vehicles are 'chased' (Canagaratna et al., 2004;Zavala et al., 2009;Zimmerman et
al., 2016), or a 'catch-all' approach in which all intercepted plumes are treated as potential exhaust
plumes (Jimenez et al., 2000;Jiang et al., 2005;Zavala et al., 2009;Hudda et al., 2013;Wang et al.,
2015). The advantage of the 'catch-all' approach is that a large number of plumes can be
encountered, leading to improved statistics for characterizing the fleet on the road in the domain of
study (Zavala et al., 2009;Wang et al., 2015). Alternatively, emission factors from mobile
measurements can be determined using a time-based or road segment-based approach in which
pollutant concentrations above background are evaluated at fixed time or distance intervals (Hudda
et al., 2013;Westerdahl et al., 2009;Zavala et al., 2006;Zavala et al., 2009). Here, we calculate fleet
emission factors using both a 'catch-all' mobile plume-based approach and a time-based approach.
**2.2.1.   Definition of background (BKG) and local (LOCAL) concentrations**
Pollutant and $CO_2$ time series were averaged to 2 s and then further smoothed using a 3
point boxcar (5 s). The background (BKG) was subsequently defined as the rolling 2nd percentile
over a 90 point (180 s) window, with additional boxcar smoothing over the same window. Similar
approaches for estimating background concentrations from mobile monitoring studies have been
employed by others (Jiang et al., 2005;Jimenez et al., 2000;Hudda et al., 2013;Park et al.,
2011;Bukowiecki et al., 2002;Larson et al., 2017). Since the background is calculated over a 3 min
window, corresponding to approximately 2 km of CRUISER travel, it is assumed to be
representative of a neighbourhood scale background (Larson et al., 2017). The "on-road"' or LOCAL
concentrations are defined as the background-corrected (i.e., above-background) mixing ratios.
Figure S2 shows sample time series for the Summer ($CO_2$, benzene, BC, HNCO, HCN) and Winter
Campaigns ($CO_2$, benzene, HNCO, HCN) and demonstrates that the LOCAL pollutant plumes were
frequently correlated with increases in $CO_2$, suggesting a combustion (i.e. vehicular) source.
**2.2.2.   Plume-based emission factor determination**
An emission factor algorithm was written using Igor Pro (Wavemetrics Inc.) to identify $CO_2$
plumes based on the first and second derivatives of the $CO_2$ time series, similar to the approach of
Wang et al. (2015). The details of the algorithm can be found in the Supplement. Briefly, the first
derivative of the $CO_2$ time series was used to identify peak boundaries and locations (peak
maxima). Two types of plumes were identified: single peak plumes (SPP) and multi-peak plumes
(MPP). Multi-peak plumes contain one or more $CO_2$ peaks (and include the SPP set). Plumes less
than 10 s in duration or with an average background-corrected $CO_2$ response of < 5 ppmv s$^{-1}$ over
the integration period were rejected as erroneous or uncaptured (Wang et al., 2015). Emission




factors (EF) expressed as mg kg-fuel$^{-1}$ for a given pollutant X and plume i were calculated using a
carbon mass balance approach:

$$EF_{X,i} = \frac{[X]}{[CO_2]} \times \frac{MW_X}{F_C \cdot MW_C} \times C_{fuel} \times 10^3 \quad [1]$$

Where [X] and [CO$_2$] are the integrated amounts of LOCAL (background-corrected) X and CO$_2$ over
the boundaries of plume $i$ in units of ppmv and ppbv respectively, MW$_X$ and MW$_C$ are the molecular
weights of pollutant X and carbon in g mol$^{-1}$, F$_C$ Is the molar ratio of carbon in CO$_2$, C$_{fuel}$ is the carbon
mass fraction in the fuel in kg C kg$_{fuel}$$^{-1}$, and 10$^3$ is the necessary unit conversion factor. A value of
C$_{fuel}$ = 0.86 was used here, which is the average of the C$_{fuel}$ for gasoline (0.85) and diesel (0.87)
(Wang et al., 2015). Strictly, the denominator in Eq. 1 should contain the sum of all emitted carbon
species (CO$_2$, CO, total hydrocarbons); however, emissions of CO$_2$ have been shown to account for >
90% of fuel consumption (Jathar et al., 2017;Yli-Tuomi et al., 2005). Plumes associated with the
highest EFs were visually inspected and in some instances were deemed to have been 'erroneously
captured' based on poor correlation between the pollutant and CO$_2$ time series; these plumes were
removed from the final dataset. Further details regarding the background calculation and peak
removal processes can be found in the Supplement.
Emission factors for benzene, toluene, C2 benzenes, NO, NO$_2$, NO$_x$, PN, and BC were obtained
during the Summer Campaign. EFs for HCN and HNCO were obtained during the Winter Campaign,
when the PICARRO measuring CO$_2$ shared the same inlet as the HR-TOF-CIMS. Statistics on the
number of plumes, plume duration, and number of peaks per plume can be found in Table S2 and
S3 at various stages of analysis for the Summer and Winter Campaigns.
### 2.2.3. Time-based emission factor determination
Emission factors were also calculated using the time-based approach, which considers the
entire data set, in contrast to the plume-based approach which only considers periods of elevated
CO$_2$ as defined by peaks (Westerdahl et al., 2009). The LOCAL (background-corrected) CO$_2$ and
pollutant mixing ratios were integrated in consecutive intervals of 30, 60, 90, and 120 s and fuel-
based emission factors were calculated according to Eq. 1.
This approach assumes that LOCAL mixing ratios are solely due to vehicle emissions (in reality,
they may also contain point sources or other types of emissions, including those not associated with
combustion). The purpose of this calculation was two-fold. First, we were interested in determining
whether this computationally simple approach could yield realistic fleet-average emission factors
comparable to those obtained using the plume-based approach. Second, we were interested in
determining EFs for pollutants not sharing a common inlet with CO$_2$ (i.e., for benzene/BC during the
winter, and HNCO/HCN during the summer), which would allow for a seasonal comparison. Here,
the assumption is that when integrating over a sufficiently long interval of time, the majority of
vehicle plumes are captured by both inlets (i.e., both the CO$_2$ and pollutant X are detected), and that
meteorology/turbulence affects the dilution of the pollutant and CO$_2$ equally.
## 3. Results and Discussion
### 3.1. Overview of mobile pollutant measurements



Ambient pollutant concentration statistics for the Summer Campaign (benzene, toluene, C2 benzenes, $NO_2$, NO, $CO_2$, PN, BC, HNCO, HCN) and Winter Campaign (benzene, HNCO, HCN, and $CO_2$) are shown in Table 1. The BC concentrations reported in this study are comparable to the range (0.10 – 1.7 µg m$^{-3}$) previously reported for Toronto (Knox et al., 2009;Chan et al., 2011).

The ambient HNCO concentrations measured during this study are similar in magnitude to those measured by others (Roberts et al., 2011;Roberts et al., 2014;Woodward-Massey et al., 2014;Zhao et al., 2014;Wentzell et al., 2013) for urban locations with minimal BB influence, which range from *ca.* 10 – 85 pptv. Recently, a much higher average summertime HNCO concentration of 1.7 ± 0.06 ppbv was measured at a suburban site in the Indo-Gangetic Plain (Kumar et al., 2018). Our measurements for both the summer and winter periods are slightly lower than the summertime mean mixing ratio of 85 pptv previously reported for a fixed location in Toronto (Wentzell et al., 2013). However we note that the authors found that HNCO was generally highest between the hours of 18:00 and 22:00. In the present study, the measurements are limited to the driving period, which could explain the slightly lower mean HNCO concentration.

The HCN mixing ratios measured in this study are two orders of magnitude lower than the mean HCN mixing ratios of 3.45 ± 3.43 ppbv (continuous sampling from a near-road location) and 1.57±0.33 ppbv (mobile measurements in heavy traffic) previously reported for Toronto (Moussa et al., 2016). Long-path FTIR measurements of HCN (1 min time resolution) were made above the busy HWY 401 in Toronto concurrent to the present study (July – Aug 2015) (You et al., 2017). Consistent with our low HCN measurements, the authors found that HCN mixing ratios only spiked above the FTIR method detection limit of 3.2 ppbv on 3 occasions (isolated 1 min data points). Although prior measurements (Moussa et al., 2016) seem exceptionally high, our measurements are also on the low end of those reported for ground level, ambient HCN in a rural region with little forest fire impact, which are on the order of a few hundred pptv (Ambrose et al., 2012). No significant long-term changes have been observed or expected for tropospheric HCN (Zhao et al., 2002) so it is unclear as to why the present measurements are so low. However, the vast majority of HCN measurements have focused on regions influenced by biomass burning and have been made aloft; measurements of HCN at ground level in urban areas are severely lacking. More measurements of HCN in urban environments are required in order to better characterize HCN concentration gradients and population exposure in regions with minimal biomass burning influence.

### 3.1.1. Local ("on-road") and background contributions: A seasonal comparison

Figure 1 shows histograms as a function of season for the measured ambient concentrations as well as for the background (BKG) and "on-road" (LOCAL) contributions for (a) HNCO and (b) HCN; Fig. S4 shows similar histograms for (a) benzene and (b) black carbon. Figure 2 shows the mean BKG and LOCAL contributions to the measured ambient concentration for the four pollutants, as a function of season. For both benzene and BC, the LOCAL contribution is dominant, indicating strong traffic sources for these pollutants. We observed a small seasonal dependence in ambient benzene, with overall higher concentrations in the winter than in the summer, as observed by others (Tan et al., 2014;Lough et al., 2005). Separation of the observations into the BKG and LOCAL contributions reveals that the shift is largely in the LOCAL contribution rather than the BKG

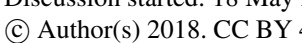



contribution, consistent with an enhanced wintertime emission factor for benzene (Tan et al.,
2014;Lough et al., 2005), attributed to higher cold-start emissions and changes in fuel composition.
An enhancement in wintertime benzene concentrations may also be partially attributed to an
increase in benzene emissions from residential wood combustion (e.g., wood stoves, fireplaces).
This enhancement would manifest in the BKG contribution (which is indeed slightly higher in the
winter than the summer). On a national scale, this source is significant (Canada-Wide Standard for
Benzene 2010 Final Report , 2012), however, in urban areas, wood heating is the primary home
heating fuel for <0.1% of residences (Matz et al., 2015), and so it is unlikely that this source is
significant within the GTA. A seasonal comparison is not available for black carbon.

10       To our knowledge, our dataset represents the first seasonal comparison of ambient HNCO
measurements made at the same location. Figure 2c illustrates a shift in HNCO concentrations from
the summer to the winter. Further inspection of Fig. 1a and 2c reveals that the seasonal difference
is largely in the BKG contribution rather than the LOCAL contribution. The lower HNCO mixing
ratios in the winter could be due to a reduction in photochemical activity and/or source strength of
secondary HNCO precursors (e.g., biogenic amines) (Woodward-Massey et al., 2014;Roberts et al.,
2014), or due to decreased influence from biomass burning. However, the extent to which wildfires
contribute to summertime HNCO concentrations is not well established and may be less significant
given HNCO's relatively short lifetime and the distant location of major Canadian wildfire events
relative to Toronto. Although residential wood burning could also contribute to HNCO across the
GTA in the winter, a recent study by Coggon et al. (2016) showed that common residential wood
fuels (e.g., heartwood and sapwood) have low nitrogen content and thus lower emissions of
nitrogen-containing VOCs such as HNCO and HCN. Consistent with this finding and the low
incidence of residential wood burning in the GTA (Matz et al., 2015), the HNCO BKG component is
low in the winter. Rather, the LOCAL component dominates the contribution to the measured HNCO
in the winter, indicating the significance of on-road emissions as an HNCO source. The similarity in
the magnitude of the LOCAL component between seasons suggests that the primary on-road HNCO
emissions remain relatively constant.

29       Similar to HNCO, we observe a strong seasonal dependence for HCN. The histogram in Fig.
1b shows a much broader distribution and higher mean for the summer compared to the winter
(Fig. 2d). Separation of the observations into BKG and LOCAL contributions in Fig. 1b reveals a
strong seasonal difference for both components, although the difference is more striking for the
BKG component. The same arguments regarding the potential impact of residential wood burning
on wintertime HNCO emissions apply to HCN. The large increase in BKG in the summer is consistent
with the wildfire season in Canada, and that biomass burning is thought to be the major source of
HCN to the atmosphere. Given the relatively long lifetime of HCN (~2-5 months) (Li et al., 2003)
compared to HNCO (~days to weeks, or even hours in clouds) (Borduas et al., 2016;Barth et al.,
2013;Zhao et al., 2014), biomass burning episodes in other parts of Canada would be expected to
have a greater potential to influence background HCN in Toronto compared to HNCO. A strong
seasonal pattern for HCN has previously been observed for tropospheric HCN column
measurements (Zhao et al., 2002); seasonal measurements of HCN at an urban location have not
been made. Unlike the other pollutants in this study, the bulk of the total measured HCN
concentration is in the BKG component rather than the LOCAL component, especially in the



summer, suggesting that in relative terms, on-road HCN sources may be less significant than other regional or global sources. Interestingly, examination of Fig. 1b also reveals a strong seasonal dependence in the LOCAL component, suggesting a possible seasonal dependence in the on-road HCN emissions, as discussed below (Sect. 3.3.3).

### 3.2. Comparison of plume-based vs time-based emission factor methodologies

A discussion of trends within the plume-based and time-based emission factors, as well as a thorough comparison of the two methodologies can be found in the Supplement for all species (Tables S6 and S7). Median EFs calculated using both the plume-based SPP approach and time-based approach (120 s interval) are also compared graphically in Fig. 3 for benzene, HNCO, and HCN. We find the time-based approach yields much higher (> 80%) median EFs for black carbon and NO than the plume-based approach. As discussed in the Supplement, the exact reason for the discrepancy is not known. However we note that both BC and NO are strongly associated with HDDV vehicles and thus exhibit highly skewed EF distributions, and that the time-based approach does not appear to adequately capture the small EF end of these distributions (Fig. S6). In contrast, we find that the two approaches yield median EFs within 25% for species associated with LDGV emissions (benzene, toluene, C2 benzenes, $NO_2$, PN, HNCO, and HCN) (see Fig. 3, Table S6, and Table S7). The ability of the time-based methodology to capture similar EF trends (Fig. S7) and magnitudes as the plume-based approach for the majority of pollutants shows that this computationally simple analysis can provide basic insight regarding fleet-average emissions, although more work is required to fully understand the conditions/pollutants which are best suited to this approach. In the current study, the advantage of the time-based methodology is its ability to reveal seasonal trends in emission factors, which are reflections of the changing LOCAL ("on-road") contributions. However, this method could potentially have useful applications for monitoring long-terms trends in vehicle emissions using near-road surveillance data or data from instruments with insufficient time resolution for a plume-based analysis.

Ultimately, periods of vehicle exhaust are defined with the highest confidence using the plume-based SPP approach and so we expect that this methodology yields the most accurate EFs. Because individual plumes are more likely to be associated with specific vehicles using this methodology, it also provides insight as to the variability of vehicle EFs and the presence of high-emitters within the fleet. Since the mean and standard deviation are sensitive to distortion by the presence of high-emitting vehicles in our modest sample sizes, we, and others (Westerdahl et al., 2009), suggest that the median and interquartile range (IQR) are more representative metrics for comparison with literature emission factors and for estimating inventories. Therefore all further discussion focuses on median EFs obtained using the plume-based SPP methodology unless stated otherwise. The distribution histograms of plume-based EFs are shown in Fig. 4 for benzene, BC, HNCO, and HCN and in Fig. S5 for others traffic pollutants (toluene, C2 benzenes, NO, $NO_2$, $NO_x$, and PN).

### 3.3. Plume-based fleet emission factors for common traffic pollutants

Our results are now compared to literature EFs for common traffic pollutants (benzene, toluene, C2 benzenes, $NO_x$, PN, BC). In the subsequent sections, the fleet-average EFs estimated for black carbon, HNCO, and HCN are discussed in further detail.

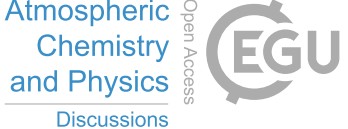

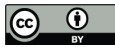

Plume-based median, mean, and interquartile range SPP EFs for a number of traffic pollutants
are listed in Table 2, along with literature EFs obtained from tunnel, mobile, near-road or remote-
sensing studies. The median/mean plume-based EFs calculated here are consistent with, but
generally fall on the lower-end of, the EFs reported in the literature. The lower EFs obtained here
may also be a result of the study location: fleet-average EFs are highly sensitive to the make-up of
the vehicle fleet (i.e., vehicle age, proportion of gasoline vs diesel vehicles, after-treatment
technologies in use), which is in turn location dependent (Kristensson et al., 2004;Zavala et al.,
2006). Furthermore, EFs from previous studies may no longer be relevant due to improvements in
emissions control technologies, removal of high-emitting vehicles, fleet-turnover, and changes in
regulations. Significant multidecadal decreases in vehicle emissions of CO, VOC, $NO_x$, PM2.5 and BC
have been observed previously (Jiang et al., 2005;McDonald et al., 2012;McDonald et al., 2013;Ban-
Weiss et al., 2008;Bishop and Stedman, 2008;Dallmann et al., 2013).
Thus, the low fleet-average EFs obtained in this study indicate that the GTA fleet is clean
relative to some of those listed for comparison in Table 2. In 1999, the government of Ontario
introduced a vehicle testing program ("Drive Clean") aimed at improving air quality by identifying
and removing/repairing high-emitting vehicles and resulting in a considerable decrease in smog-
causing pollutants ($NO_x$ and total hydrocarbons) of about 16% from program inception to 2010
(McCarter, 2012). Coincident with changes in gasoline regulations for benzene and other
technology improvements, Canada introduced a Canada-Wide Standard for Benzene in 2010. Since
then, the transportation sector has led a dramatic reduction in national average ambient
concentrations of benzene, particularly in urban locations (Canada-Wide Standard for Benzene
2010 Final Report , 2012). Our relatively low EFs are hence consistent with the successful
implementation of these and other policies, such as reduction in fuel sulfur content.
The most recent emission factor measurements for comparison were made at a near-road
location in Toronto in 2013/2014 (Wang et al., 2015). The mean and median EFs for the VOCs, $NO_x$,
PN, and BC obtained here are in excellent agreement with those reported by Wang et al. (2015) (see
Fig. 4 and Fig. S5). The approach for determining EFs presented here differs in a) its mobile nature,
covering a wide geographical area and range of road-types and b) its short time period (i.e., limited
number of captured plumes). However, the mobile nature of our study results in a higher likelihood
of sampling exhaust from a larger spectrum of vehicle types (including HDDVs) under a greater
range of real-world driving conditions. The good agreement between the two studies for a wide
range of pollutants gives confidence that our methodology provides representative fleet-average
emission factors despite a smaller sample size. In this way the current study compliments the
stationary study (Wang et al., 2015), demonstrating that the EFs obtained at their fixed location are
applicable across a large region.
### 3.3.1. Black carbon emission factors
We obtained plume-based median and mean black carbon emission factors of 24.9 mg $kg_{fuel}^{-1}$
and 85.6 mg $kg_{fuel}^{-1}$ respectively (IQR: 10.3 – 76.4 mg $kg_{fuel}^{-1}$). BC emission factors obtained by
prior tunnel, near-road, and mobile studies are listed in Table S9 for comparison. Literature
emission factors for HD diesel vehicles range from 160 – 2400 mg $kg_{fuel}^{-1}$, one to two orders of
magnitude higher than the literature emission factors for LD gasoline vehicles, which range from


~1 – 300 mg kg$_{fuel}$$^{-1}$. LDGV emission factors at the high end of this range are from older studies in more polluted environments (Westerdahl et al., 2009). In many of the earlier studies, BC emission factors were obtained using aethelometers with a 1 min time resolution, which may not have been fast enough to accurately quantify BC emission factors. In the current study, the poor performance of the time-based approach with respect to yielding BC EFs in agreement with the plume-based approach may also indicate that high time-resolution measurements of BC and good plume definition are required to accurately estimate BC EFs from mobile measurements. However, more comparisons are needed to determine if and how calculated BC EFs depend upon the BC measurement method.

As was observed for the other pollutants (benzene, NO$_x$, PN etc.), the BC emission factors obtained in this study are on the lower end of the reported literature range. Two studies have made recent measurements of the mixed vehicle fleet in Toronto. Wang et al. (2015) made BC EF measurements from their near-road stationary site in downtown Toronto using a photoacoustic soot photometer and report a mean EF of (35 – 55) mg kg$_{fuel}$$^{-1}$. Liggio et al. (2012) obtained BC EFs from transect driving downwind, and perpendicular to, a major Toronto highway (mean HDDV fraction ~3.3%). The authors report fleet-average median emission factors of 59.3 mg kg$_{fuel}$$^{-1}$ (IQR: 27.0 – 148.4 mg kg$_{fuel}$$^{-1}$) using a HS-LII instrument and 29.4 mg kg$_{fuel}$$^{-1}$ (IQR: 11.8 – 66.0 mg kg$_{fuel}$$^{-1}$) using a single-particle soot photometer. The values from these two studies (Wang et al., 2015;Liggio et al., 2012) lie between the median and mean obtained in the current study.

The lower values obtained in this study compared to Liggio et al. could be reflective of overall changes in the vehicle fleet over the past 5 years leading to reductions in BC emissions, consistent with observations at other locations (Ban-Weiss et al., 2008;Dallmann et al., 2012). The discrepancy between our study and the other two Toronto studies could also be related to location: their fixed/limited sites may not be representative of the full fleet across the GTA. Given the difference in LDGV and HDDV BC EFs, the emission factor calculation will be quite sensitive to the frequency at which each vehicle type is sampled, which will be location-dependent. This sensitivity can be quite dramatic: a recent study (Dallmann et al., 2013) found that due to their higher associated BC emissions, even a small fraction (<1%) of heavy duty trucks can significantly bias the calculated LDGV emission factors (by over 40%). For a pollutant exhibiting wide inter- and intra-vehicle variation in EFs, obtaining measurements that capture the full fleet make-up over a range of driving conditions is critical. Although we did not record the number of HDDVs (expected fraction ~ 4%), the mobile design and scope of our study helps to mitigate location-specific results. Overall, we found that the top 4% of plumes had vehicle emissions greater than 320 mg kg$_{fuel}$$^{-1}$ which are typical of heavy-duty vehicles.

### 3.3.2. HNCO emission factors

As previously mentioned, literature HNCO EFs have been obtained exclusively from a limited number of dynamometer studies (on both gasoline and diesel vehicles) and so a comprehensive understanding of the real-world magnitude and variability of HNCO EFs is lacking. Here we obtain the first fleet-average EFs for HNCO. Table 3 compares the HNCO emission factors available in the literature with the wintertime plume-based HNCO median EF obtained in this study (2.3 mg kg$_{fuel}$$^{-1}$).



Our time-based analysis (Fig. 3b) suggests that HNCO EFs are similar in the summer and winter
(with slightly higher EFs in the summer, contrary to the behaviour of benzene).
Only two previous dynamometer studies (Suarez-Bertoa and Astorga, 2016;Brady et al., 2014)
obtained HNCO emission factors from gasoline vehicles and the average EFs reported from those
studies differ by more than an order of magnitude. The plume-based median EF obtained in this
study is about a factor of two higher than that obtained by the earlier study (fleet average of
0.91±0.58 mg $kg_{fuel}^{-1}$ for 8 LDGV) (Brady et al., 2014), but significantly lower than that obtained
more recently (fleet average of 93 mg $kg_{fuel}^{-1}$ for 3 LDGV, or 29 mg $kg_{fuel}^{-1}$ if the anomalously high
LDGV is omitted) (Suarez-Bertoa and Astorga, 2016).
Interestingly, emission factors ranging from 0.21 – 3.96 mg $kg_{fuel}^{-1}$ were recently obtained from
an engine dynamometer study on a single light duty diesel engine, in agreement with the results
from the current study (Wentzell et al., 2013). This may suggest that HNCO emissions from gasoline
and diesel vehicles are of similar magnitude. In contrast, HNCO emission factors for an off-road
diesel engine have been found to be an order of magnitude higher, and it has been suggested that
the magnitude and range of HNCO emissions, as well as their dependence on operating conditions,
could be different for this type of engine (larger, off-road diesel engine) (Link et al., 2016;Jathar et
al., 2017). Much of the early work on HNCO vehicle emissions was prompted by the finding that
selective-catalytic reduction (SCR) systems could constitute an important source of HNCO (Kröcher
et al., 2005;Heeb et al., 2012;Heeb et al., 2011), but the impact of SCR systems (or other control
technologies such as the diesel particulate filter, DPF, or diesel oxidation catalyst, DOC) is disputed
(Jathar et al., 2017).
In addition to a wide range of emission factors, the available literature revealed conflicting
information on the conditions leading to elevated HNCO emissions, as well as high inter-vehicle
variability. HNCO emissions have been observed to vary by as much as an order of magnitude
depending on the driving cycle, but the influence of hard acceleration and cold engine starting is
contested (Brady et al., 2014;Suarez-Bertoa and Astorga, 2016). Similarly, studies have
demonstrated opposite trends for idle vs active operating conditions (Link et al., 2016;Wentzell et
al., 2013). For all these reasons, a direct comparison of the EF obtained in this study to reported EFs
is challenging. The current study cannot reveal the mechanism of HNCO production from diesel or
gasoline vehicles, or its dependence on factors such as driving condition and the presence of
various after-treatment technologies. However, a key strength of our study is that it is based upon a
large number of vehicles operating on-road in real-world conditions, thus implicitly reflecting a
range of these factors. Therefore, we suggest that the IQR reported here (1.37 – 4.15 mg $kg_{fuel}^{-1}$)
along with the overall distribution of measured HCNO EFs (Fig. 4c) provides the most realistic
constraint to date on the magnitude and variability of HNCO emissions.

### 3.3.3. HCN emission factors

As with HNCO, EFs for HCN have been obtained exclusively from a limited number of
dynamometer studies. Table 4 lists HCN emission factors obtained in this study along with those
obtained from prior dynamometer studies; here distance-based units (mg km⁻¹) are used for ease of
comparison. The seasonal dependence of HCN vehicle emissions has not been previously studied.



Interestingly, Fig. 3c shows that the HCN EFs obtained using the time-based approach exhibit a
strong seasonal dependence, with the median summertime EF almost a factor 5 higher than the
median wintertime EF. This behaviour is opposite that of benzene, which has higher wintertime EFs
by about a factor of 2 due to enhanced cold-start emissions and changes in gasoline composition
(Lough et al., 2005). Although the mechanisms for HCN and benzene formation are different (the
former involving chemistry on high temperature catalysts), the reasons for the higher HCN EFs in
the summer are not known.
Early studies on some of the first generation three-way catalysts yielded very high HCN
emission factors, typically under abnormal or malfunctioning operating conditions (Bradow and
Stump, 1977;Keirns and Holt, 1978;Cadle et al., 1979;Urban and Garbe, 1979, 1980). The magnitude
of the HCN emissions exhibited high car-to-car variability and a strong dependence on operating
condition, as well as the presence and composition of the catalysts. An average LDGV HCN EF of
12.1 mg km$^{-1}$ was estimated from a review (Harvey et al., 1983) of these early studies – over two
orders of magnitude greater than the EFs obtained here. However, those EF estimations were
obtained for all driving modes for both normal and abnormal operating conditions.
Given the significant improvements in catalyst and emissions reduction technologies since the
1970s and 1980s, the applicability of these early studies to current HCN emission is questionable.
Certainly, more recent studies (Karlsson, 2004;Baum et al., 2006;Becker et al., 1999;Moussa et al.,
2016) suggest that present-day HCN EFs are much lower with individual vehicle EFs ranging from 0
– 11.7 mg km$^{-1}$ (see Table 4). However, these limited dynamometer studies also reveal a large inter-
vehicle variability in HCN EFs, with no clear pattern between emissions and vehicle characteristics
(e.g. age). The most recent study (Moussa et al., 2016) also showed that intra-vehicle EFs are highly
sensitive to fuel injection technology (e.g. gasoline direct injection, GDI vs. port-fuel injection, PFI),
after-treatment technology (presence and absence of a particulate filter), and operating conditions
(e.g., aggressiveness of driving cycle, hot vs. cold-starts).
The median winter HCN EF obtained in this study (using either the plume-based or time-based
approach) is over an order of magnitude lower than the average EF obtained by the most recent
dynamometer study (Moussa et al., 2016). The higher summer HCN EF obtained by the time-based
analysis is in better agreement, although it is still low. However, due to the aforementioned
variability in the dynamometer results, a direct comparison is not straightforward. As with HNCO,
our study provides the most comprehensive HCN emission factors available to date since the
mobile design allows us to obtain EFs for a large number of vehicles, thereby capturing the real-
world inter- and intra-vehicle variability of emissions. Similarly, the IQR (0.32 – 0.88 mg kg$_{fuel}$$^{-1}$) and
distribution of measured EFs (Fig. 4d) give new insight into the range of on-road HCN emission
factors.
**3.4. Emission factor distributions: contributions from high-emitters**
The spread in the EFs for all measured pollutants is wide, consistent with prior mobile studies
(Park et al., 2011;Hudda et al., 2013;Zavala et al., 2009).  Such variability is expected given the
differences in speed, acceleration, grade, and inter-vehicle variability occurring on-road. As
illustrated by Fig. 4 and Fig. S5, the EFs are log normally distributed, with the degree of skewness



dependent on pollutant. Skewness in EF distributions is typically attributed to the presence of 'high-emitting' vehicles among the fleet, but may also arise from the range and transient nature of driving conditions experienced in the real-world (e.g., hard acceleration). The distributions provide insight into the strategy for emission reductions. From a policy perspective, pollutants exhibiting a more normal distribution may be most effectively targeted by tightening fleet-wide regulations while those exhibiting a more skewed distribution may be most effectively targeted, initially, by the removal of high-emitters (Hudda et al., 2013).

Cumulative emission factor distributions for several pollutants are presented in Fig. 5. These plots highlight the relative skewness of EFs for each pollutant by displaying the fraction of total emissions as a function of the fraction of vehicles, sorted from largest to smallest EF. The distributions are highly skewed for NO and PN, and exceptionally skewed for BC, as observed by others (Jiang et al., 2005;Hudda et al., 2013;Liggio et al., 2012). This behaviour is expected given that these pollutants are emitted in large quantities from diesel-powered vehicles, which represent a small fraction of the fleet (Jiang et al., 2005;Ban-Weiss et al., 2008;Jimenez et al., 2000;Dallmann et al., 2012;Liggio et al., 2012;Wang et al., 2015;Ban-Weiss et al., 2010;Dallmann et al., 2013;Tan et al., 2014) and hence were encountered by CRUISER less often. For NO, it is also likely that an unknown quantity of emitted NO is being converted to $NO_2$ before plume capture (hence the high frequency of EFs in the lowest bin, < 0.15 g $kg_{fuel}^{-1}$), further exacerbating the skewness. For BC, the top 25% worst emitters, likely all diesel vehicles, contribute to more than 80% of the total emissions, while the top 5% contribute to almost 50%. At a near-road site in Toronto, the top 25% worst emitters were found to contribute to 100% of the total BC emissions, with the top 5% contributing > 60% (Wang et al., 2015). As more heavy-duty vehicles become equipped with particulate filters and advanced $NO_x$ abatement technologies (i.e., SCR systems) the overall EF distributions for pollutants such as BC and NO may shift, but the skewness could actually increase unless high-emitters, such as the older, legacy diesel vehicles, are specifically targeted (McDonald et al., 2013).

The EF distributions for the VOCs were less skewed (Jiang et al., 2005;Hudda et al., 2013;Wang et al., 2015). The benzene, HNCO, and HCN profiles in Fig. 5 are similar, with the top 25% worst emitters contributing to 55-60% of the total emissions and the top 5% contributing 20-30%. The less skewed distributions for HNCO and HCN may indicate that their HDDV EFs are not significantly higher than their corresponding LDGV EFs. The least skewed pollutant in this study is $NO_2$ – the top 25% worst emitters only contribute to ~50% of total emissions and the top 5% contribute to ~15%. As suggested above, post-tailpipe conversion of NO to $NO_2$ is likely occurring prior to measurement. The cumulative emission factor distribution for $NO_x$ (=NO + $NO_2$) more closely resembles the distribution for VOCs and $NO_2$ than NO.

### 3.5. Vehicle emission estimates for Canada

Annual emissions for Ontario and Canada can be estimated using the fuel-based EFs and from annual sales of gasoline and diesel. The assumption here is that the gasoline and diesel sales are proportional to the number of gasoline- and diesel-powered vehicles on the road, and that the EFs obtained from the mobile measurements reflect this distribution. A summary of total vehicle emissions of $NO_x$, benzene, BC, HNCO, and HCN calculated using the median plume-based emission



factors are given in Table 5. Nationwide inventory estimates for $NO_x$ (Air Pollutants Emissions
Inventory), benzene (Canada-Wide Standard for Benzene 2010 Final Report, 2012), and BC
emissions (Canada's Black Carbon Inventory: 2017 Edition, 2017) by the transportation sector are
also listed in Table 5 for comparison. For all three pollutants, the scaled up emissions were more
than a factor of 2 lower than the inventory estimates. Using the mean EFs rather than median
reduces this discrepancy. Our results suggest that the inventories may be overestimated but more
work is required to understand the reasons for the difference.
We estimate that on a national scale, 104 tonnes of HNCO and 24 tonnes of HCN are emitted
annually by on-road vehicles. These values are lower than the recent nationwide estimates of 250 –
770 tonnes HNCO for 2010 (Wentzell et al., 2013) and 703 tonnes HCN for 2012 (Moussa et al.,
2016), owing to the lower fleet-average EFs obtained in this study. These vehicle emissions can be
placed in the context of their respective biomass burning emissions. Total wildfire emissions of CO
during the 2015 wildfire season (May 31, 2015 – Nov 2, 2015) were calculated using FireWork-
GEM-MACH (Pavlovic et al., 2016). These total CO emissions were then scaled by literature
emission ratios (ER) expressed as mol of pollutant per mol of CO to estimate biomass burning
emissions (Table 5). Only a few studies have investigated HNCO ERs (Veres et al., 2010;Roberts et
al., 2011). Biomass burning emissions of HCN have been the subject of a greater number of studies,
but a recent review notes that the HCN/CO ER can be different for different fire types and that even
within single or similar fire types there is a high variability in HCN emissions (Akagi et al., 2011).
In 2015, HNCO emissions from forest fires were estimated at 5377 tonnes and 40 tonnes for
Canada and Ontario respectively. Although on a national scale the HNCO vehicle emissions are over
an order of magnitude lower than the biomass burning emissions, in urban areas the vehicle source
becomes relatively more significant. This is seen in the provincial comparison, where the greater
population density and lower frequency of forest fires in Ontario results in HNCO vehicle emissions
comparable in magnitude to biomass burning emissions. When secondary formation of HNCO from
precursors in vehicle exhaust is also taken into account (Link et al., 2016;Liggio et al., 2017b), the
significance of vehicle emissions as a source of HNCO will likely be further enhanced.
In 2015, the HCN emissions from forest fires were estimated at $(1.2-5.8) \times 10^4$ tonnes and
(87-431) tonnes for Canada and Ontario respectively. At the national scale, the biomass burning
emissions are about 3 orders of magnitude greater than the vehicle emissions. Even at the
provincial scale, the biomass burning emissions are about an order of magnitude greater than the
vehicle emissions. This result is consistent with the large BKG component to the ambient
measurements made in the study. If the summertime EF obtained using the time-based approach is
used (2.7 mg $kg_{fuel}^{-1}$) then the total vehicle emissions of HCN are estimated at 125 tonnes and 44
tonnes for Canada and Ontario respectively, still lower than previous estimates (Moussa et al.,
2016). Although biomass burning emissions continue to be the dominant source of HCN in this
estimation, the potential significance of vehicles as a source of HCN, especially in urban areas with
minimal BB influence, is non-negligible.
**4.  Conclusions and Implications**





We deployed a mobile laboratory over a large metropolitan area, capturing exhaust emissions
from a large number of vehicles under a range of operating conditions and driving environments.
Plume-based and time-based algorithms were developed to estimate EFs from the on-road
measurements. The plume-based method avoids cumbersome cross-reference with recorded
vehicle plumes (i.e, as in 'vehicle chase' methods) and shows potential for obtaining real-world EFs
from limited-term mobile studies with minimal computational effort. The time-based method was
found to perform well for pollutants with less skewed EF distributions (i.e., not associated with high
HDDV emissions), and best for pollutants with minimal local sources (i.e., HNCO and HCN). Further
studies are required to fully validate the time-based method, but this approach could potentially be
used to calculate EFs from near-road sites with lower time-resolution datasets. Both methodologies
could thus be efficient ways of rapidly monitoring trends in emission factors, especially for
pollutants whose emissions are likely to be influenced by emerging technologies or policies. Hence,
this approach could be valuable for documenting accountability.
Based on good agreement of the plume-based EFs with reported literature EFs for common
traffic pollutants, and the more precise definition of vehicle exhaust for this methodology, the
plume-based EFs are considered to be superior to the time-based EFs. Due to the broad range of
vehicles and real-world conditions captured by the measurements, the plume-based algorithm
applied to the mobile study provides a better average EF for use in scaling-up emissions or for
assessing general exposure than a limited number of dynamometer studies. We thereby obtain the
first, and most representative fleet-average emission factors for HNCO and HCN, and insight into
their real-world variability.
The plume-based EF obtained for black carbon in this study (median: 25 mg $kg_{fuel}^{-1}$, IQR: 10
– 76 mg $kg_{fuel}^{-1}$) is consistent with decreases in vehicular BC emissions over time (Ban-Weiss et al.,
2008;Dallmann et al., 2013). Despite this improvement, our work, like that of others, shows that a
small number of vehicles (predominantly HDDV) are responsible for a disproportionate amount of
the on-road BC emissions. As a result, BC concentrations, and hence exposure, are highest near
highways and major roadways, and efforts to target these emissions will likely have a strong impact
on local air quality. In North America, GDI vehicles are replacing PFI vehicles, which currently
dominate the light-duty fleet (Chan et al., 2014). GDI vehicles promise advantages such as lower
fuel consumption, but have been shown in recent studies to emit more BC than their PFI
counterparts (Saliba et al., 2017) – although introduction of gasoline particulate filters could
mitigate this effect (Chan et al., 2014;Saliba et al., 2017). Similarly future decreases in diesel
emission of BC are predicted (Dallmann et al., 2012) as the fleet turns over and more diesel trucks
on the road are equipped with diesel particulate filters. Therefore, it is critical that fleet emissions
of BC are monitored in the future, with careful attention to the relative contributions from heavy-
duty vs light-duty vehicles. Since BC also impacts global climate change (Highwood and Kinnersley,
2006;Bond et al., 2013), mitigating vehicle emissions of BC has the dual benefit of meeting air
pollution and climate targets (Bahadur et al., 2011;Bond et al., 2013).
Overall, our results indicate that a vehicle fleet dominated by light duty gasoline vehicles is a
source of HNCO and HCN to the atmosphere, with plume-based median EFs under wintertime, 'real-
world', driving conditions of 2.3 mg $kg_{fuel}^{-1}$ (IQR: 1.4 – 4.2 mg $kg_{fuel}^{-1}$) and 0.52 mg $kg_{fuel}^{-1}$ (IQR: 0.32 –




0.88 mg kg$_{fuel}^{-1}$) respectively. Given our poor understanding of how emerging emission control
technologies (e.g., SCR systems, diesel oxidation catalysts) influence HNCO emissions, it is
imperative that fleet emissions of HNCO are studied over time. The impact of vehicle emissions on
secondary HNCO production in urban areas should also be investigated.
Our work demonstrates that HCN emission factors obtained in out-dated dynamometer studies
for LDGVs equipped with first-generation three-way-catalysts under abnormal operating conditions
(Harvey et al., 1983) are not applicable to the present day. However, they indicate that the most
recent dynamometer studies (Moussa et al., 2016;Karlsson, 2004) may also overestimate real-
world HCN emissions. Overall, the relatively small vehicle emission factor obtained in this study
suggests that vehicles are not likely a significant source of HCN on a regional and larger scale.
However, in view of the discrepancies between this study and others (Moussa et al., 2016), and the
paucity of HCN measurements in urban locations, more work is required to establish the
atmospheric significance of vehicle emissions of HCN at the neighbourhood and smaller scale. In
particular, the extent and cause of variation in HCN concentrations and emission factors, which
appear to vary widely in ambient measurements and dynamometer studies respectively, should be
further constrained and understood. Future research should also seek to understand the reasons
for the observed seasonal variation in HCN concentrations and emission factors.
**Supplement**
Supplementary material related to this article is available.
**Acknowledgements**
We thank the technical support staff and information management/information technology team of
AQRD for assistance with equipment and data system installation, data management, and driving.
We thank Amy Leithead for assistance with the PTR-TOF-MS and Junhua Zhang for providing the
wildfire CO estimates from Firework-GEM-MACH. This program was supported by the Clean Air
Regulatory Agenda (CARA).
**Competing Interests**
The authors declare that they have no conflict of interest.

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



1 **Figures**

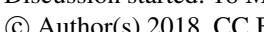

**Figure 1.** Distribution of ambient mixing ratios for (a) HNCO and (b) HCN. Top panel: total
concentration. Middle panel: background (BKG) concentration. Bottom panel: background-
corrected (LOCAL) concentration. Summer Campaign (July 2015) shown as colored bars, Winter
Campaign (Jan 2016) shown as grey bars.


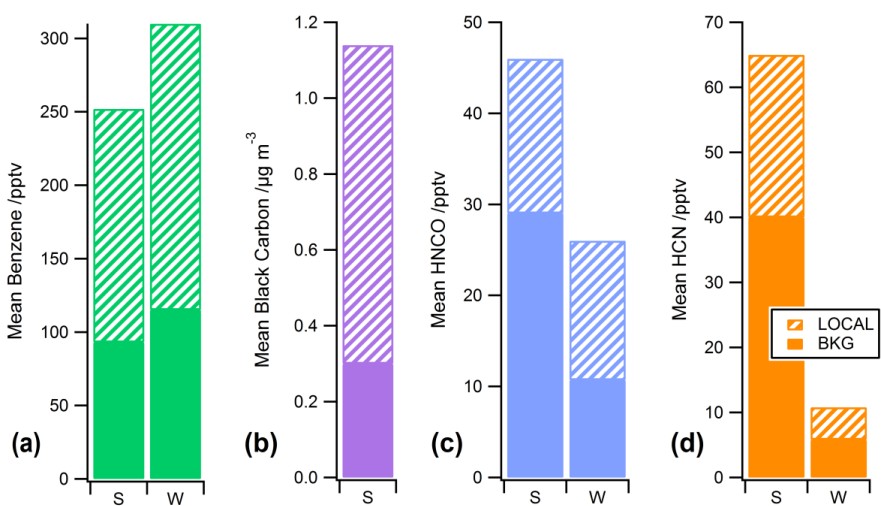

**Figure 2.** Mean on-road (LOCAL, patterned) and background (BCK, solid) mixing ratios for the
Summer (S) and Winter (W) Campaigns. (a) benzene, (b) BC, (c) HNCO, and (d) HCN.

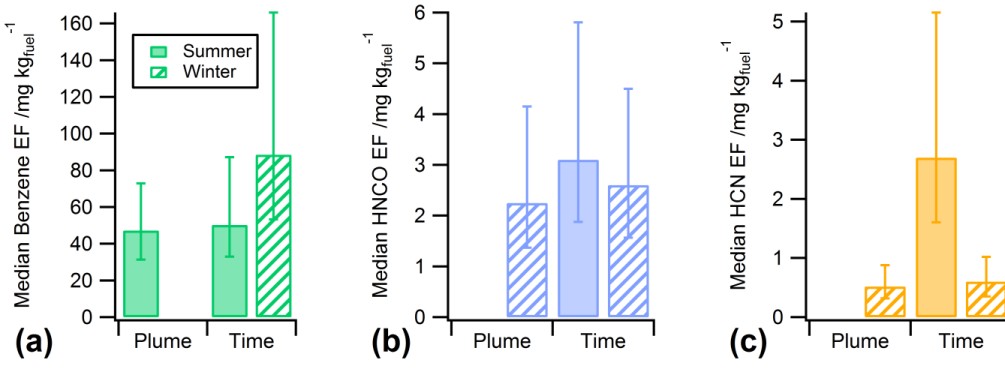

**Figure 3.** Median emission factors for (a) benzene, (b) HNCO, and (c) HCN calculated using the SPP
plume-based approach (Plume) or the time-based approach with an integration period of 120 s
(Time). The error bars show the interquartile range. Values obtained from the Summer Campaign
(solid bars) and Winter Campaign (patterned bars).




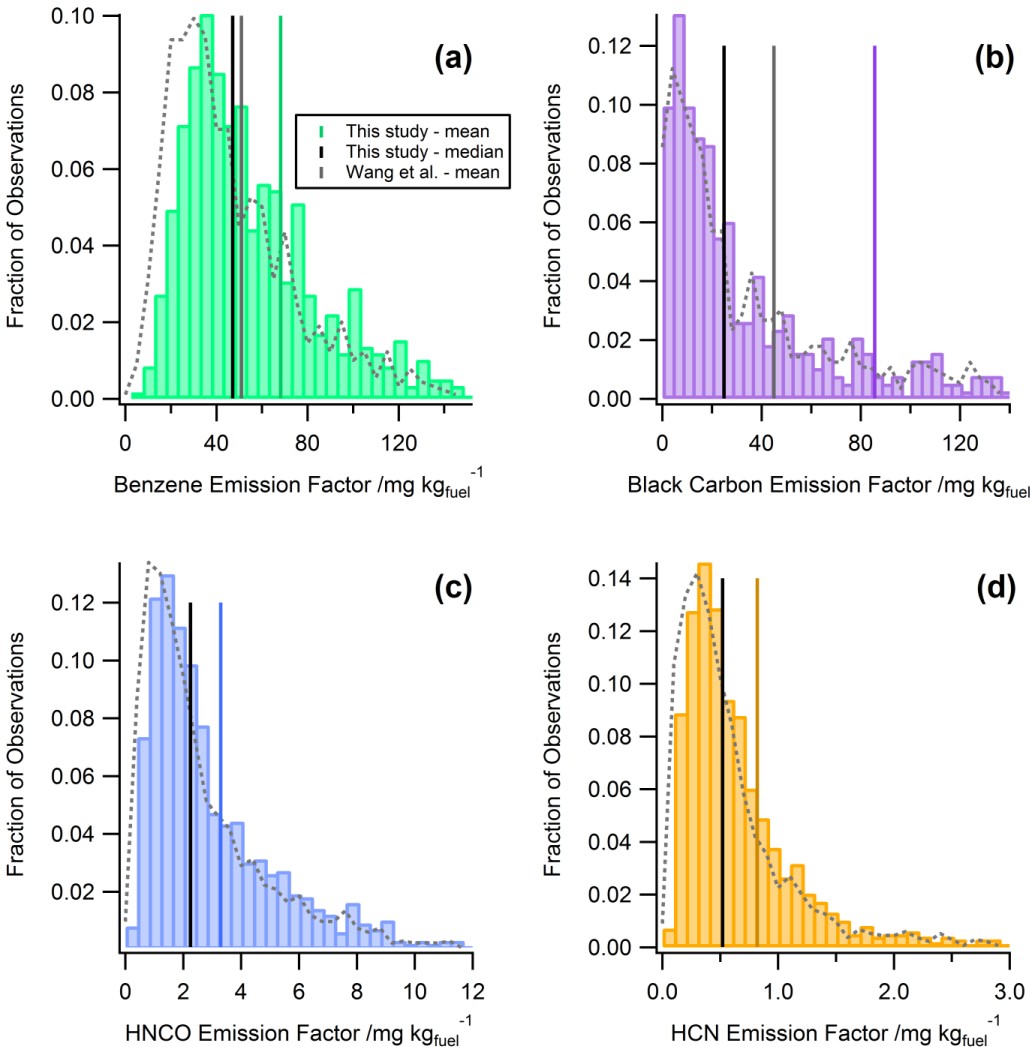

Figure 4. Plume-based emission factors obtained by CRUISER for (a) benzene, (b) BC, (c) HNCO, and (d) HCN for the SPP case (colored bars) and the MPP case (grey, dashed line). The median and mean EF values are indicated by the vertical black and colored lines respectively. Where available, the mean EF obtained by Wang et al. (2015) is indicated by the vertical grey line.





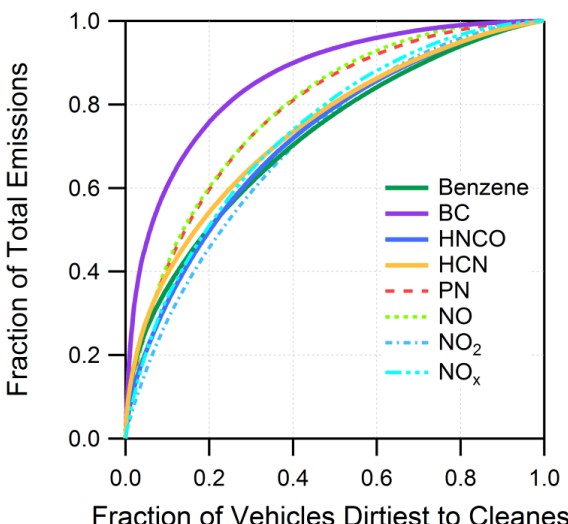

**Figure 5.** Cumulative emission factor distributions for SPP plume-based measurements: benzene
(solid, green), BC (solid, purple), HNCO (solid, blue), HCN (solid, yellow), NO (dotted, light green),
$NO_2$ (dotted, light blue), $NO_x$ (dotted, light teal), particle number (dotted, pink). A 1 to 1 line would
indicate that all vehicles have the same emission factor.





1 **Tables**

2 **Table 1.** Method of detection and ambient concentration statistics for selected pollutants on

3 CRUISER

| Pollutant and Units | Instrument | N | Mean (1σ) | Median | 25th Percentile | 75th Percentile | Max |
|---|---|---|---|---|---|---|---|
| **Summer Campaign** | | | | | | | |
| $C_6H_6$ /pptv | PTR-TOF-MS | 112156 | 293 (1043) | 170 | 91 | 320 | 170000 |
| $C_7H_8$ /pptv | PTR-TOF-MS | 113760 | 913 (3210) | 382 | 198 | 757 | 395525 |
| $C_8H_{10}$ /pptv | PTR-TOF-MS | 112156 | 605 (224) | 237 | 123 | 468 | 191205 |
| $NO_2$ /ppbv | LGR[a] | 99041 | 12.7 (13.3) | 8.5 | 4.2 | 17.5 | 552 |
| HNCO /pptv | HR-TOF-CIMS | 103951 | 45.1 (39.0) | 39.3 | 28.4 | 53.8 | 2168 |
| HCN /pptv | HR-TOF-CIMS | 103951 | 63.8 (52.4) | 56.6 | 41.0 | 75.4 | 2429 |
| NO /ppbv | TECO (42iTL)[b] | 98036 | 20.9 (39.7) | 6.2 | 2.0 | 21.2 | 998 |
| $CO_2$ /ppmv | PICARRO[c] | 59275 | 415 (33) | 408 | 392 | 435 | 1893 |
| $PN^f$ /1000# $cm^{-3}$ | CPC[d] | 98888 | 42.0 (87.4) | 24.7 | 15.5 | 45.6 | 9230 |
| Black Carbon /µg $m^{-3}$ | HS-LII[e] | 106075 | 1.06 (2.32) | 0.38 | 0.16 | 0.97 | 43.4 |
| **Winter Campaign** | | | | | | | |
| $C_6H_6$ /pptv | PTR-TOF-MS | 131882 | 315 (294) | 262 | 169 | 388 | 18500 |
| HNCO /pptv | HR-TOF-CIMS | 119642 | 25.7 (54.7) | 15.5 | 8.8 | 27.1 | 2985 |
| HCN /pptv | HR-TOF-CIMS | 119642 | 10.6 (15.7) | 7.7 | 5.4 | 11.2 | 1579 |
| CO2 /ppmv | PICARRO[c] | 78683 | 439 (30) | 408 | 419 | 449 | 1250 |

Principles of Operation: [a]Cavity-enhanced laser absorption spectroscopy, [b]Thermo Scientific (42iTL) Chemiluminescence, [c]Cavity ring-down spectroscopy, [d]Light scattering, [e]Laser-induced incandescence. [f]PN=Ultrafine Particle Number Counts. Statistics obtained after self-sampling algorithm was applied to the high-time resolution data with N data points. All instruments operated at 1s resolution except PICARRO (2 s). The mean daily temperature was *ca.* 25 °C during the Summer Campaign and *ca.* -5 °C during the Winter Campaign.



**Table 2.** Plume-based median and mean emission factors calculated using single-peak plumes
(SPP) for the Summer and Winter Campaigns. Interquartile range (25th – 75th percentile) shown in
brackets. Units for numerator given in the pollutant column, units for denominator given in the
header.

| Pollutant and Units | Fuel-based units $/kg_{fuel}^{-1}$ | Distance-based units[a] $/km^{-1}$ | Literature Range Fuel-based units[b] $/kg_{fuel}^{-1}$ | References |
|---|---|---|---|---|
| **SUMMER** | | | | |
| Benzene /mg | **47.2, 68.2** (31.3 - 72.8) | 3.7, 5.7 | **28 - 650** | Gentner et al., 2013; Hwa et al., 2002; Wang et al., 2015; Araizaga et al., 2013; Ho et al., 2009; Zavala et al., 2009; Kristensson et al., 2004 |
| Toluene /mg | **101.6, 179.5** (62.5 - 194.6) | 8.4, 14.9 | **50 - 2075** | Hwa et al.,2002; Gentner et al., 2013; Wang et al., 2015; Araizaga et al., 2013; Ho et al. 2009; Zavala et al., 2009; Kristensson et al., 2004 |
| C2 Benzenes[c] /mg | **76.8, 147.6** (44.8 - 149.7) | 6.4, 12.2 | **74 - 1455** | Hwa et al.,2002; Gentner et al., 2013; Wang et al., 2015; Araizaga et al., 2013; Ho et al. 2009; Zavala et al., 2009; Kristensson et al., 2004 |
| $NO_2$[d] /g | **1.15, 1.39** (0.56 - 1.85) | 0.095, 0.115 | | |
| NO[d] /g | **1.03, 1.79** (0.38 - 2.20) | 0.086, 0.148 | | |
| $NO_x$ (=NO + $NO_2$) /g | **2.27, 3.13** (1.16 - 4.23) | 0.188, 0.259 | **1.4 - 42** | Wang et al., 2015; Kristensson et al., 2004; Hwa et al., 2002; Jiang et al., 2005; Hudda et al., 2013, Park et al., 2011; Dallmann et al., 2013; Kirchstetter et al., 1999; Ban-Weiss et al. 2008 |
| Particle Counts /$10^{14}$ # | **8.3, 15.9** (3.7 - 20.0) | 0.69, 1.32 | **3.9 - 57.4** | Wang et al., 2015; Kristensson et al., 2004; Hudda et al., 2013; Ban-Weiss et al., 2010 |
| Black Carbon /mg | **24.9, 85.6** (10.3 - 76.4) | 2.1, 7.1 | **10 - 2400** | Literature Comparison in Table S9 |
| **WINTER** | | | | |
| HNCO /mg | **2.25, 3.30** (1.37 - 4.15) | 0.126, 0.274 | | Literature Comparison in Table 5 |
| HCN /mg | **0.52, 0.82** (0.32 - 0.88) | 0.043, 0.068 | | Literature Comparison in Table 6 |

[a]Conversion from fuel-based units to distance-based units based on a fleet composed of 96% LDV with a fuel consumption rate of 10.6 L/100 km and 4% HDV (vehicles > 4.5 tonnes) with a fuel consumption rate of 28.5 L/100 km (based on the 2009 Canadian Vehicle Survey, and combining MDV with HDV) (Natural Resources Canada, 2012). Fuel densities at 15 °C of 730 kg m$^{-3}$ (gasoline/LDV) and 840 kg m$^{-3}$ (diesel/HDV) were used in all cases.

[b]For EFs reported in distance-based units, conversion to fuel-based units using stated distribution of gasoline and diesel vehicles and fuel consumption rates where available. When not stated, fuel consumption rates of 10.6 L/100 km for gasoline vehicles and 33.4 L/100 km for diesel vehicles were used (based on 2009 Canadian Vehicle Survey) (Natural Resources Canada, 2012). If the distribution of vehicles in the study was not stated or unclear the conversion was done assuming 96% gasoline and 4% diesel. Fuel densities at 15 °C of 730 kg m$^{-3}$ (gasoline/LDV) and 840 kg m$^{-3}$ (diesel/HDV) were used in all cases.

[c]C2 Benzenes corresponds to the sum of m-, p-, and o-xylene and ethylbenzene (protonated formula C8H11+). For literature reporting EFs for the individual species, the individual EFs were summed together.

[d]Comparison to literature made for $NO_x$ and not NO or $NO_2$ due to a) unknown conversion of NO to $NO_2$ post-tailpipe in our study and b) reporting of $NO_x$ rather than NO or $NO_2$ in the literature.



1 **Table 3.** Comparison of literature HNCO emission factors from the exhaust of various gasoline and
2 diesel fueled engines in fuel-based units (mg $kg_{fuel}^{-1}$).

| Reference | Type of study | HNCO detection | Range /mg $kg_{fuel}^{-1}$ | Average /mg $kg_{fuel}^{-1}$ | Description of vehicle and fuel |
|---|---|---|---|---|---|
| This study | Mobile | HR-TOF-CIMS | 1.4 – 4.2[a] | 2.3, 3.3[b] 2.6, 4.0 3.1, 5.4 | Winter fleet, plume-based (SPP) Winter fleet, time-based (120 s) Summer fleet, time-based (120 s) |
| Wentzell et al., 2013 | Engine Dynamometer | Acetate-TOF-CIMS | 0.21-3.96 | NA | 2011 Jetta equipped with turbo diesel injection (TDI) and diesel oxidation catalyst (DOC) |
| Brady et al., 2014 | Chassis Dynamometer | Acetate-TOF-CIMS | 0.45 – 1.70 (fleet averages for the 4 phases) | 0.91±0.58 (full fleet, entire drive cycle) | 8 LDGVs equipped with three way catalyst (TWC) |
| Suarez-Bertoa & Astorga, 2016 | Chassis Dynamometer | FTIR | NA | 30 (23 °C)[c] 140 (-7 °C)[c] 93 (23 °C)[d] 29 (23 °C)[e] | 10 LDVs: 3 LDGV, 4 LDDV, 2 flex-fuel LDV, 1 electric LDV. Varying after-treatment |
| Heeb et al., 2011 | Engine Dynamometer | Offline LC-MS analysis, after derivatization | NA | 29 (with combined DPF-SCR system) 32 (with $V_2O_5$-based SCR system) | Diesel engine with a turbo charger and direct fuel engine, with and without selective catalytic reduction (SCR) and without diesel particulate filter (DPF) |
| Jathar et al., 2017 | Engine Dynamometer | Acetate-TOF-CIMS | 31 – 56 | NA | John Deere PowerTech Plus (off-road) diesel engine with DOC and DPF, with and without SCR; diesel and biodiesel |
| Link et al., 2016 | Engine Dynamometer | Acetate-TOF-CIMS | NA | 54±3 (Idle) 17±2 (50% Load) | Same engine as above, with no DOC, DPF or SRC; diesel and biodiesel |

[a]Interquartile range
[b]Median, Mean
[c]Fleet median for all 10 vehicles (all other values in the paper are reported in distance-based units mg $km^{-1}$)
[d]Mean for the 3 gasoline vehicles (LDGVs)
[e]Mean for the gasoline vehicles omitting GV3 (anomalously high EFs)



**Table 4.** Comparison of literature HCN emission factors from the exhaust of various gasoline and diesel fueled engines in distance-based units (mg km$^{-1}$).

| Reference | Type of Study | HCN detection | Range /mg km$^{-1}$ | Average /mg km$^{-1}$ | Description of vehicles and fuel |
|---|---|---|---|---|---|
| This study | Mobile | HR-TOF-CIMS | 0.03-0.07[a] | 0.043, 0.068[b]<br>0.046, 0.069<br>0.21, 0.37 | Winter fleet, plume-based (SPP)<br>Winter fleet, time-based (120 s)<br>Summer fleet, time-based (120 s) |
| Bradow and Stump, 1977 | Chassis and Engine Dynamometer | Offline after trapping by NaOH | <LOD (normal operation)<br>0.0 – 75.6 (malfunctioning) | NA | 3 LDGVs w/ TWC (1977)<br>5 LDGVs w/o TWC (1976) |
| Keirns and Holt, 1978 | Chassis Dynamometer | Offline after trapping by NaOH | < 1.4 (LOD) (normal operation)<br>0.8 – 11.8 (malfunctioning) | NA | 1 LDGV w/ and w/o TWC of varying composition (1977) |
| Cadle et al., 1979 | Chassis Dynamometer | Trapping by NaOH with colorimetric detection | 0-14.4 | 6.9 (no catalyst)<br>0.6 (oxidation catalyst)<br>3.1 (dual or three-way catalyst)<br>8.1 (rich malfunction with TWC) | 26 LDGVs (production and experimental, 1967-1978) |
| Urban and Garbe, 1979 | Chassis Dynamometer | Trapping by NaOH, GC-ECD | 0.0 – 2.4 (normal)<br>0.3 - 2.3 (malfunctioning) | 0.2 (normal, excluding LDV w/o catalyst) | 5 LDGVs (1977-1978), 1 w/o catalyst, 4 w/ oxidation catalyst |
| Urban and Garbe, 1980 | Chassis Dynamometer | Trapping by NaOH, GC-ECD | 0.1 – 1.1 (normal)<br>0.0 – 112.3 (malfunctioning) | | 4 LDGVs with TWC (1978-1979) |
| Harvey et al., 1983 | Review | NA | 1.0 – 12.1 (weighted normal and malfunctioning averages for LDVs w/ different catalyst cases) | 7.1 | 206 LDVs (non-catalyst, oxidation catalyst, TWC), 11 HDVs, gasoline and diesel |
| Becker et al., 1999 | Chassis Dynamometer | FTIR | NA | < 2 (below LOD) | 21 LDGVs (1996-1997) |
| Karlsson, 2004 | Chassis Dynamometer | Trapping by NaOH with colorimetric detection | 0.0 – 11.7 | 2.2 ±4.2 | 5 LDGVs (1989-1998) |
| Moussa et al., 2016 | Chassis Dynamometer | PTR-TOF-MS | 0.0 – 5.6 | 1.4±1.7 | 3 LDGV (2008-2011) |

[a]Interquartile range
[b]Median, Mean



1 **Table 5.** Annual traffic pollutant emissions from the transportation sector and biomass burning for
2 Canada and Ontario

| | | NOx | Benzene | BC | HNCO | HCN |
|---|---|---|---|---|---|---|
| Canada 2015 | Vehicle Emissions[a] (tonnes) | $1.05 \times 10^5$ | 2180 | 1150 | 104 | 24 |
| | Forest Fires[b] (tonnes) | | | | 5377[c] | $(1.2^d - 5.8^e) \times 10^4$ |
| | Vehicle Emissions Inventory Estimates (tonnes) | $4.26 \times 10^{5f}$ | 6600[g] | 6401[h] | | |
| Ontario 2015 | Vehicle Emissions[a] (tonnes) | $5.29 \times 10^4$ | 775 | 1410 | 37 | 9 |
| | Forest Fires[b] (tonnes) | | | | 40[c] | 87[d]-431[e] |

[a]In 2015, net sales for gasoline and diesel in Canada were $4.26 \times 10^{10}$ L and $1.80 \times 10^{10}$ L respectively (Statistics Canada). The total mass of fuel is calculated assuming a fuel density at 15 °C of 730 kg m$^{-3}$ and 840 kg m$^{-3}$ for gasoline and diesel respectively, for a nationwide total of $4.62 \times 10^7$ tonnes of fuel. In 2015, net sales of gasoline and diesel in Ontario were $1.63 \times 10^{10}$ L and 5.43 × 109 L respectively, for a provincial total of $1.64 \times 10^7$ tonnes (Statistics Canada). Vehicle emissions calculated using the plume-based SPP emission factors.
[b]Wildfire CO emissions were calculated to be 5003 ktonnes and 37.3 ktonnes for Canada and Ontario, respectively
[c]HNCO emission calculated using HNCO/CO ER for Oak Woodland of 0.7 mmol HNCO mol CO$^{-1}$ (Veres et al., 2010)
[d]HCN emission calculated using HNCO/CO ER of 0.00242 mol HCN mol CO$^{-1}$ (Rinsland et al., 2007)
[e]HCN emission calculated using HNCO/CO ER of 0.012 mol HCN mol CO$^{-1}$ (Akagi et al., 2011)
[f]Estimated NO$_x$ emissions for 2015 from light and heavy duty diesel and gasoline vehicles, trucks and motorcycles (Air Pollution Emission Inventory: https://pollution-waste.canada.ca/air-emission-inventory)
[g]Estimated on-road transportation benzene emissions for 2008 (Canada Wide Standard for Benzene: 2010 Final Report)
[h]Estimated BC emissions for 2015 from diesel (5679 tonnes) and gasoline (722 tonnes) on-road vehicles (Canada's Black Carbon Inventory: 2017 Edition)

5

