# Peer review of "Elucidating real-world vehicle emission factors from mobile measurements over a large metropolitan region: a focus on isocyanic"

_Atmospheric Chemistry and Physics, 2018_

## Referee Comment (RC1) · Anonymous Referee #1 · 6 Aug 2018

General Comments: The paper by Wren et al. focuses on quantifying HNCO, HCN and BC emission factors using mobile summer and wintertime measurements conducted over 9 days (in July 2015) and 8 days (in Jan 2016), respectively in the Greater Toronto Area.

In general, the paper is well written and the work and results are quite interesting and will improve current understanding of concerning the traffic emission source of HNCO and HCN to the atmosphere. I recommend publication in ACP after the following comments have been addressed.

[Figure]

Comments: Abstract: Line 20-23: "Our results demonstrate that although biomass burning is a dominant source of both air toxics on a national scale, vehicular emissions play an increasingly important role at a local scale, especially in heavily-trafficked urban areas." This statement is not really a new scientific result but rather something that is expected to hold generically, so am not sure you need it in the abstract in the absence of quantitative information concerning the national and local scale emissions here.

Introduction: line 6-9: "However, it is not established if these species are directly responsible for negative outcomes associated with TRAP, or if they act in tandem with, or as proxies for, other compounds in the pollutant mixture (Brook et al.,2007;Mauderly and Samet, 2009;Dominici et al., 2010)." Please clarify: I don't think there is doubt about direct health impact of CO and NO. Page 4; Line1: While Roberts et al. did indeed calculate the concentration exposure of documented concern, they relied on Wang et al. for the toxicological basis so it makes sense to also cite Wang et al. 2007. Citation: Wang, Z., et al., 2007. Protein carbamylation links inflammation, smoking, uremia and atherogenesis. Nat. Med. 13 (10), 1176–1184.

Page 6; Line 1-2: The description of the algorithm for excluding self-sampling could be given in a few lines here and the reader can be referred to the supplement for details as this is an important issue. Section 2.1.2and Section 2.1.3: The technical description of the PTR-TOF-MS and HR-TOF-CIMS is too sketchy in the main manuscript and warrant some more description. The supplement does have the details so atleast the operational parameters (Townsend ratio, humidity dependent sensitivity reported as a range and correction magnitude, detection limits and number of samples below detection limit) can be added here in 3-4 lines.

Comment and suggestion: The PTR-TOF-MS can also measure HNCO and it would be very useful for readers to know how the HR-TOF-CIMS using the iodide ionization method measurements and the PTR-TOF-MS measurements of HNCO compare? As reported by Kumar et al., 2018 in Scientific Reports, which the authors cite in another context, some of the amide and amine precursors of HNCO can also be detected using the PTR-TOF-MS. This data would provide more insights and help improve the subsequent discussion of HNCO sources and in my view should be included in the revised manuscript. 2.1.4. High-sensitivity laser-induced incandescence (HS-LII) for Black Carbon: What is known about its performance Vs the traditional BC measurement devices like an aethalometer? Are there inter-comparisons that are available in the literature?

Considering that the emission factors reported in this study are much lower than many other studies for HCN, it is important to discuss the possibility of systematic "biases" intrinsic to the techniques that may have been used to quantify HNCO, HCN and BC in other studies to avoid pinning down the contribution of such effects on other factors and ambient variability alone.

Page 7; Line 21: How much would the results change if instead of 2nd percentile one used 5th or 10 th percentile? It important to provide the range of final values resulting from such choices. Page 7 ; Line 31: Please include the r values for these correlations. Page 8; Lines 16-17: In the daytime the photochemical transformation of NO does pose a problem for such analysis. How much time would have typically elapsed between emission from the tail pipe and its sampling and measurement. What were the ozone mixing ratio in summer and winter? How often did it rain during the sampling days? Page 9; Line 2: Why are toluene and C2- benzenes missing from the winter campaign list? Page 9; Line 9: Wasn't it a suburban site influenced strongly by open agricultural BB fires? Page 9; Line 14: Is it possible to construct a 9 to 17 hours diurnal variation plot from the composite data for HNCO and HCN. This would throw light on the photochemical source strength for HNCO? Page 9; Line 16: "The HCN mixing ratios measured in this study are two orders of magnitude lower than the mean HCN mixing ratios of 3.45 ± 3.43 ppbv. The standard deviation does not reflect ambient variability for a dataset that is normally distributed. Can the authors clarify?

Page 10; Lines 16-19: What is the HNCO lifetime in the atmospheric environment of Toronto? Actually unless there is strong wet scavenging it can be quite long lived since

it reacts very slowly with hydroxyl radicals, if at all.

Secondly it is not clear to me as to why there should be less BB in winter. In most places, BB emissions and resultant pollutant concentrations peak during winter due to increased emission (people burn more BB to keep warm in winter if they do not have access to cleaner energy sources) and there is suppressed mixing due to the lower boundary layer in winter. So the authors should clarify and improve this discussion.

Page 12; Lines 5-8: Looking at the road sampling sites which are located near the coast, what could be the effect of humid sea-land breezes during the day on the measured emission factors, as CO2 and other more soluble gases like HNCO could experience different dilution/removal effects. Can these effects of the sea-land breeze be completely ignored? Page 14; Lines 305: Was the HNCO measurement technique identical in all these studies? Please clarify.

Conclusions: This section is very well written and some of the quantitative findings reported here can be put in the abstract which currently has scope for improvement. Figure 2: BKG instead of BCK in the Figure caption?

Supplement: Page 2: The molecular formula of tricholorobenzene is incorrect. Should be C6 instead of C3 in the molecular formula. Also given the low mixing rations observed after background correction, the authors should explain the magnitude and methodology for the sensitivity corrections owing to changing humidity of sample air in more detail.
* * *

---

## Referee Comment (RC2) · Anonymous Referee #2 · 10 Aug 2018

Wren et al. present emissions findings from a short mobile laboratory deployment in the Greater Toronto Area (GTA) in the summer and winter of 2015-2016. Their manuscript results mostly focus on trends in ambient concentrations and emissions factors of black carbon (BC), isocyanic acid (HNCO), and hydrogen cyanide (HCN). A key finding of this study is that emission factors of BC, HNCO, and HCN are lower than those reported in the literature suggesting that the mobile source fleet in GTA is relatively clean. They also report that mobile sources contribute substantially to the ambient concentrations of HNCO and HCN in urban environments and may be more important than biomass

burning in those regions.

The methods used, the presentation of results, and the conclusions based on the results are robust. The manuscript is very well written and easy to follow; amongst the top 10 percentile of manuscripts I have reviewed. I have very few comments on the manuscript (see below) and strongly recommend publication in ACP. Despite being limited to GTA, the methodology and results from this manuscript will be very useful to the air quality and atmospheric chemistry community.

1. Page 5, lines 36-38: The inlet appears to be quite long given that HNCO and HCN – that tend to be sticky molecules – might suffer large losses. Were the losses through this length and this tubing material quantified for HNCO and HCN? What are the implications of tube losses on the study? Also, if the material can stick and be released as and when the equilibrium between the material on the tube and in the tube is perturbed, would the measured delay result in miscalculations?

2. Page 5, line 37: Easy to calculate but what was the residence time in the sampling line?

3. Page 6, line 20: When mentioning the supplement, can you specify the correct section in the supplement?

4. Page 9, line 9: Include findings about agricultural burning from Chandra and Sinha (2016).

5. The seasonal differences described in Section 3.1.1 and visualized in Figure 2 may not be directly interpretable based on differences in the absolute concentrations in the two seasons. For example, for the same emissions and sources, wintertime concentrations for a species can be higher simply from shallower boundary layer heights. So higher wintertime concentrations may not reflect changes in emissions or sources. On the same note, for the same source, ambient temperatures may result in very different emissions, e.g., Suarez-Bertoa et al.(2016) found vehicular emissions of HNCO to be

much higher at lower temperatures. Furthermore, for the same source and emissions, photochemistry could influence the background concentrations and contributions to the total. Would it have been better to compare seasonal differences after ratio-ing the species of interest against an inert tracer such as CO?

6. Isn't the statement starting on page 10, line 42 about LOCAL versus BKG HCN also true for HNCO?

7. Section 3.3: Am I understanding this right that the emission factors are calculated only using the LOCAL estimates? Also, if the LOCAL estimates include emissions from near-road non-mobile sources, the emission factors in this work would serve as an upper bound?

8. It isn't clear to me how one would go about doing this but given the different pollutants measured and differences in their relative proportions in gasoline versus diesel vehicles., there must be a way to apportion the various pollutants measured into their contributions from gasoline and diesel. This would be an interesting exercise with huge value to air quality managers/regulators.

9. Section 3.3.3: Why was HCN compared in Table 6 in mg/km while HNCO was compared in Table 5 in mg/kg-fuel?

10. Assuming that the BKG estimates are representative of ambient concentrations away from the roadway, would it be safe to say that the HNCO concentrations in GTA are significantly lower than the 1 ppbv health threshold suggested by Roberts et al. (2011).

---

## Author Comment (AC1) · 21 Sep 2018

**Elucidating real-world vehicle emission factors from mobile measurements over a large metropolitan region: a focus on isocyanic acid, hydrogen cyanide, and black carbon**

Sumi N. Wren, John Liggio, Yuemei Han, Katherine Hayden, Gang Lu, Cris M. Mihele, Richard L. Mittermeier, Craig Stroud, Jeremy J. B. Wentzell, Jeffrey R. Brook

Air Quality Process Research Section, Air Quality Research Division, Environment and Climate Change Canada, 4905 Dufferin St., Toronto, ON, M3H 5T4

Corresponding Authors: Sumi N. Wren (sumi.wren@gmail.com), Jeffrey R. Brook (jeff.brook@canada.ca)

**Response to reviewer comments:**

The authors thank the reviewers for the excellent comments and recommendations. In the following, the reviewer comments are in bold text followed by our response in normal (non-bold) text. Modified text taken from the manuscript is italicized.

**Referee #1**

**General Comments: The paper by Wren et al. focuses on quantifying HNCO, HCN and BC emission factors using mobile summer and wintertime measurements conducted over 9 days (in July 2015) and 8 days (in Jan 2016), respectively in the Greater Toronto Area. In general, the paper is well written and the work and results are quite interesting and will improve current understanding of concerning the traffic emission source of HNCO and HCN to the atmosphere. I recommend publication in ACP after the following comments have been addressed.**

**Comments:**
**Abstract: Line 20-23: "Our results demonstrate that although biomass burning is a dominant source of both air toxics on a national scale, vehicular emissions play an increasingly important role at a local scale, especially in heavily-trafficked urban areas." This statement is not really a new scientific result but rather something that is expected to hold generically, so am not sure you need it in the abstract in the absence of quantitative information concerning the national and local scale emissions here.**

This line was removed.

**Introduction: line 6-9: "However, it is not established if these species are directly responsible for negative outcomes associated with TRAP, or if they act in tandem with, or as proxies for, other compounds in the pollutant mixture (Brook et al.,2007;Mauderly and Samet, 2009;Dominici et al., 2010)." Please clarify: I don't think there is doubt about direct health impact of CO and NO.**

We have changed the wording of the sentence slightly to change the emphasis: "However, it is not established *to what extent* these species are solely responsible for negative outcomes associated with TRAP, or *to what degree* they act in tandem with, or as proxies for, other compounds in the

pollutant mixture." This acknowledges the health impacts of CO and NOx, but suggests that there is more to it. As discussed in Brook et al. 2007 – NOx could be acting as an indicator for some other pollutant exposure affecting a population (e.g. surrogate for other pollutant(s) originating from motor vehicles or high-temperature combustion, such as volatile organic compounds (VOCs) or polycyclic aromatic hydrocarbons). We have added a sentence to this effect:

*"That is, $NO_x$ could be an indicator for other pollutants originating from vehicular combustion, including volatile organic compounds and particulate species (Brook et al., 2007)."*

**Page 4; Line1: While Roberts et al. did indeed calculate the concentration exposure of documented concern, they relied on Wang et al. for the toxicological basis so it makes sense to also cite Wang et al. 2007. Citation: Wang, Z., et al., 2007. Protein carbamylation links inflammation, smoking, uremia and atherogenesis. Nat. Med. 13 (10), 1176–1184.**

The reference has been added.

**Page 6; Line 1-2: The description of the algorithm for excluding self-sampling could be given in a few lines here and the reader can be referred to the supplement for details as this is an important issue.**

A few lines describing the self-plume exclusion algorithm have been added in this section:

*"Briefly, the self-sampling algorithm identified periods of 'potential' exhaust based on CRUISER speed, relative wind speed, and wind direction (towards inlet) and periods of 'suspected' exhaust within these windows, based on the presence of exhaust tracers (BC, NO, fine particle counts). Periods of 'suspected' exhaust were removed from the data."*

**Section 2.1.2 and Section 2.1.3: The technical description of the PTR-TOF-MS and HR-TOF-CIMS is too sketchy in the main manuscript and warrant some more description. The supplement does have the details so at least the operational parameters (Townsend ratio, humidity dependent sensitivity reported as a range and correction magnitude, detection limits and number of samples below detection limit) can be added here in 3-4 lines.**

Thank you, we agree. A few lines have been added in the main manuscript to both sections to better describe the instruments.

Section 2.1.2: *"The PTR-TOF-MS was operated with an E/N value of 140 Td. Air for analysis by the PTR-TOF-MS was sampled off the common gas phase inlet via a 2 m long PFA tube with 0.52 cm ID at a rate of 4.4 sLpm and the instrument sampled part of this flow (100 sccm) through a 120 cm insulated PEEK capillary with 0.08 cm ID heated to 70°C. Mass spectra were acquired with a time resolution of 1 s and a resulting mass resolution of approx. 4000 m/Δm. The response of the PTR-TOF-MS to specific VOCs was determined using a home-built zero/calibration unit and a custom VOC gas standard (Ionicon). The 2σ detection limits differed slightly for the Summer and Winter Campaigns and were calculated respectively to be 110 pptv and 155 pptv for benzene, 125 pptv and 240 pptv for toluene, and 110 pptv and 160 pptv for C8 benzenes. The sensitivities and detection limits are also listed in Table S1."*

Section 2.1.3: *"Air for analysis was drawn at ~22 sLpm through a 3 m long heated (50 °C) inlet (0.58 cm ID). The CIMS subsampled from this flow into the molecule reaction (IMR) region via a critical orifice at 1.7 sLpm. Mass spectra were acquired with a time resolution of 1 s and a resulting mass*

*resolution of approx. 5000 m/Δm. Calibrations of HNCO were conducted by thermally decomposing cyanuric acid at 250 °C to HNCO (Roberts et al., 2010) with the permeation rate quantified via Fourier Transform Infrared Spectroscopy (FTIR; Thermo-Fisher Inc.). Calibrations of HCN were performed by diluting a HCN gas standard (Air Liquide, ppmv in $N_2$) in zero air.  Humidity dependant response factors for both species were derived by diluting the calibration gas flows with humidified air to a final RH ranging from ~9% to 90%, resulting in sensitivities of 0.086 ncps/pptv and 0.1 ncps/pptv for HCN and HNCO respectively.  The 2σ detection limits for HNCO and HCN were estimated to be 7 pptv each for both the Summer and Winter Campaigns."*

**Comment and suggestion: The PTR-TOF-MS can also measure HNCO and it would be very useful for readers to know how the HR-TOF-CIMS using the iodide ionization method measurements and the PTR-TOF-MS measurements of HNCO compare? As reported by Kumar et al., 2018 in Scientific Reports, which the authors cite in another context, some of the amide and amine precursors of HNCO can also be detected using the PTR-TOF-MS. This data would provide more insights and help improve the subsequent discussion of HNCO sources and in my view should be included in the revised manuscript.**

We agree that this would make a very valuable addition to the manuscript. The PTR-TOF mass spectra were fit including masses corresponding to protonated HNCO, protonated HCN, and several protonated amides (formamide, methylformamide, methylacetamide/propanamide). However, the PTR-TOF-MS was not very sensitive during this study and the response was generally in the noise at these m/z. HCN also has a weak proton affinity and thus detection by PTR-TOF-MS is a less ideal approach (HCN detection with a proton transfer reaction mass spectrometer, Knighton et al., Int. J. Mass Spectrom., 283, 112-121, 2009). Furthermore, we did not calibrate the PTR-TOF-MS for HNCO, HCN, or for precursor amides (nor did we determine the humidity dependent sensitivities). Therefore, performing such an analysis is beyond the scope of this study.

**2.1.4. High-sensitivity laser-induced incandescence (HS-LII) for Black Carbon: What is known about its performance Vs the traditional BC measurement devices like an aethalometer? Are there inter-comparisons that are available in the literature?**

The HS-LII used in this study is a new prototype developed by Atrium Technologies Inc. and the National Research Council (NRC). The performance of a previous HS-LII instrument relative to a single-particle soot photometer (SP2) was briefly addressed in a previous study by our group ("Are emission of black carbon from gasoline vehicles underestimated? Insights from near and on-road measurements", Liggio et al., Environ. Sci. Technol., 46, 2012). We found that SP2 black carbon measurements are biased low due to the inability of this instrument to measure particles with a diameter < 70 nm (the HS-LII does not have this limitation since ensemble properties are measured). Unpublished data shows that the two instruments are in good agreement on a time-resolved basis. In another paper ("Time-resolved measurements of black carbon light absorption enhancement in urban and near-urban locations of southern Ontario, Canada", *Atmos. Chem. Phys.*, 11, 2011) concurrent measurements of BC particles with a photoacoustic spectrometer (PA) and a previous HS-LII demonstrated that the PA measurement is sensitive to the presence and amount of non-refractory material (coating) while the HS-LII measurement is significantly less-sensitive, thus yielding a "truer" BC mass concentration. Additional text on the performance of the HS-LII has been added to the Method Section 2.1.4: "*An advantage of this technique is that it determines ensemble properties for all particles within the sample volume and so does not suffer from a particle size limitation; previous studies have shown that the HS-LII can detect laboratory generated particles < 7 nm in diameter (Stirn et al., 2009). As a result, a previous study found that BC measurements by a*

*single-particle soot photometer (SP2), which is only sensitive to particles with a diameter > 70 nm, are biased low relative to the HS-LII (Liggio et al., 2012). Furthermore, it has been shown that the HS-LII is significantly less-influenced by the presence of non-refractory mass compared to other BC measurements methods such as photoacoustic spectrometers (Chan et al., 2011)."*

We also note that in Section 3.3.1 (first paragraph on black carbon emission factors) we have already noted: "However, more comparisons are needed to determine if and how calculated BC EFs depend upon the BC measurement method." We are not aware of any intercomparisons with an aethelometer.

**Considering that the emission factors reported in this study are much lower than many other studies for HCN, it is important to discuss the possibility of systematic "biases" intrinsic to the techniques that may have been used to quantify HNCO, HCN and BC in other studies to avoid pinning down the contribution of such effects on other factors and ambient variability alone.**

There are no obvious systematic biases that would reduce HCN concentrations significantly. The relative uncertainties in the HCN measurements which are used to derive the HCN emission factors are approximately 30%. This cannot sufficiently account for the large disparity between the EFs derived here and in the literature. Other systematic biases which could reduce HCN EFs may include the potential for liquid water within the sampling system, inaccurate background subtraction (ie: systematically high), and unknown ion chemistry within the CIMS. Liquid water condensed on the sampling lines would result in the dissolution of HCN and ultimately drive concentrations (and EFs) downwards. While this cannot be ruled out entirely, the fact that the sampling line was externally heated (see methods) suggests that this was not likely. Background subtraction has been carefully assessed (see methods, supplement and response to comment below) and is also not likely to reduce EFs significantly. Finally, unknown CIMS ion chemistry could potentially result in decomposition of HCN with the ion-molecule region, although no obvious evidence of such chemistry has been observed.

**Page 7; Line 21: How much would the results change if instead of 2nd percentile one used 5th or 10th percentile? It important to provide the range of final values resulting from such choices.**

The sensitivity of the results to changes in the definition of the background (percentile and boxcar size) are briefly addressed in the Supplement (section 1.5). An increase in the percentile from the 2nd to the 5th percentile was found to reduce the median benzene EF by approx. 7%. Since our approach for defining the background is consistent with what others have done, the influence of changing parameters in the background definition was not performed methodically for all other pollutants. We would also like to note here that Wang et al. (2015) performed extensive sensitivity tests on the influence of various plume definitions and a line about this has been added to the Supplement (section 1.5):

*"We note that Wang et al. (2015) performed extensive sensitivity tests on the influence of plume definition on mean EFs."*

A line has been added towards the end of Section 2.2.2 (Plume-based emission factor determination) to direct readers towards the Supplement:

*"Further details regarding the background calculation and their influence on calculated EFs, and the peak removal processes can be found in the Supplement (Section 1.5)."*

**Page 7; Line 31: Please include the r values for these correlations.**

The correlation between the pollutants was not calculated. Visual inspection shows that they frequently co-vary. The wording has been changed so as not to be misleading: "Figure S2 shows sample time series for the Summer ($CO_2$, benzene, BC, HNCO, HCN) and Winter Campaigns ($CO_2$, benzene, HNCO, HCN) and demonstrates that the LOCAL pollutant plumes ***frequently co-varied*** with increases in $CO_2$, suggesting a combustion (i.e. vehicular) source."

**Page 8; Lines 16-17: In the daytime the photochemical transformation of NO does pose a problem for such analysis. How much time would have typically elapsed between emission from the tail pipe and its sampling and measurement. What were the ozone mixing ratio in summer and winter? How often did it rain during the sampling days?**

Given that the mobile platform was traversing the roads, and the average width of the single-peak plumes was 20-25 s, we expect the time from emission for NO to be quite short (on the order of minutes). It is of course not possible to exactly determine the time from emission, and so the potential for reactions of NO with ozone cannot be ruled out. We have therefore added text to the manuscript here to note the potential for ozone reactions:

"*We note that for NO, reactions with ozone can result in a low bias for NO EFs. In this study we expect the time from emission to be on the order of minutes, although exact emission times are not known. As such it is likely that the EF for NO here represent lower limits to the true NO EFs.*"

The mean ozone mixing ratio in the summer (driving times only) was approx. 30 ppbv ($25^{th}$ percentile = 14 ppbv, $75^{th}$ percentile = 45 ppbv). Winter ozone data was not available. It only rained on one day (July 17) during the Summer Campaign. It snowed one day during the Winter Campaign (January 18) and there was snow/slush on the ground during the campaign.

**Page 9; Line 2: Why are toluene and C2- benzenes missing from the winter campaign list?**

Although toluene and C2-benzenes (and the other pollutants) were measured in the winter, the focus of this paper is the vehicle emission factors. A plume-based analysis was only available for toluene and C2-benzenes (and the other pollutants) in summer because $CO_2$ was only sampled from the common gas phase inlet during that season. Benzene was addressed for both seasons since it is a well-known traffic pollutant with a more constrained EF. In this study it helped to a) validate the EF methodologies and b) contrast the behavior of BC, HNCO and HCN. For these reasons, we reported benzene, but not toluene and C2-benzenes, for the winter.

**Page 9; Line 9: Wasn't it a suburban site influenced strongly by open agricultural BB fires?**

This section has been changed to add the results of Chandra and Sinha (2016) and to better represent these results:

"*However, Chandra and Sinha (2016) report annual HNCO mixing ratios of 0.94 ppbv for a suburban site in the Indo-Gangetic Plain that is strongly influenced by crop-residue fires; a much higher average summertime HNCO concentration of 1.7 ± 0.06 ppbv was recently measured at the same site (Kumar et al., 2018).*"

**Page 9; Line 14: Is it possible to construct a 9 to 17 hours diurnal variation plot from the composite data for HNCO and HCN. This would throw light on the photochemical source strength for HNCO?**

While it is possible to construct a partial diurnal profile from the data, the resultant plots are not useful. This is essentially because we were mostly confined to driving at the same midafternoon 2-5 hour time period, resulting in insufficient data at other times to be used in this manner.

**Page 9; Line 16: "The HCN mixing ratios measured in this study are two orders of magnitude lower than the mean HCN mixing ratios of 3.45 _ 3.43 ppbv. The standard deviation does not reflect ambient variability for a dataset that is normally distributed. Can the authors clarify?**

Because the measurements are being made on the road and involve measurements close to the background as well as very high measurements due to vehicle emissions (spiking as high as > 1 ppb) the standard deviation does not simply reflect ambient variability. The mean and standard deviation listed in Table 1 was calculated from the 1 s data (no smoothing or averaging applied). We also note that Table 1 contains values for the median, 25th and 75th percentile, and maximum value which provide greater insight into the distribution of the ambient measurements.

**Page 10; Lines 16-19: What is the HNCO lifetime in the atmospheric environment of Toronto? Actually unless there is strong wet scavenging it can be quite long lived since it reacts very slowly with hydroxyl radicals, if at all.**

Yes, the lifetime of HNCO with respect to photolysis or reaction with OH is on the order of months to hundreds of years. In clouds it can be reduced (Borduas et al., 2016; Zhao et al.,2014; Barth et al., 2013), with the lifetime dependent on cloud liquid water content and pH (on the order of days/week). It only rained on one day (July 17) during the Summer Campaign. It snowed one day during the Winter Campaign (January 18) and there was snow/slush on the ground during the campaign. Here we meant the shorter lifetime of HNCO compared to HCN (with respect to distant BB episodes having impact in Toronto). The wording has been changed and the lifetime range for HNCO has been added:

"*However, the extent to which wild fires contribute to summertime HNCO concentrations is not well established and may be less significant given HNCO's moderate lifetime (as short as a few hours in clouds, but typically weeks to hundreds of years) (Borduas et al., 2016;Barth et al., 2013;Zhao et al., 2014), and the distant location of major Canadian wildfire events relative to Toronto.* "

**Secondly it is not clear to me as to why there should be less BB in winter. In most places, BB emissions and resultant pollutant concentrations peak during winter due to increased emission (people burn more BB to keep warm in winter if they do not have access to cleaner energy sources) and there is suppressed mixing due to the lower boundary layer in winter. So the authors should clarify and improve this discussion.**

This was already addressed in this section. Biomass burning is not common in Toronto in the winter as a source of heat. Furthermore, suppressed mixing would be expected to lead to higher HNCO mixing ratios, all else being equal, and that is not what was observed. The potential impact of a lower boundary layer in the winter has been added in these seasonal discussions (Section 3.1.1 paragraphs on benzene, HNCO, and HCN respectively):

*"Increases in wintertime benzene may also be attributed to a shallower boundary layer height."; "Shallower boundary layer heights would be expected to lead to enhanced wintertime concentrations if HNCO emissions/sources remain constant, and yet we observe lower concentrations of HNCO in the winter. Hence the boundary layer height is a potential issue that can reduce the apparent differences between summer and winter""; "As with HNCO, despite shallower boundary layer heights in the winter, the overall concentrations are observed to be lower in the winter."*

**Page 12; Lines 5-8: Looking at the road sampling sites which are located near the coast, what could be the effect of humid sea-land breezes during the day on the measured emission factors, as CO2 and other more soluble gases like HNCO could experience different dilution/removal effects. Can these effects of the sea-land breeze be completely ignored?**

Since EFs are essentially a ratio of these soluble gases, provided that dilution acts upon both species similarly then such effects are not relevant. Although we did observe increases in relative humidity while passing through lake air, the timescales in this study are too short for the gases to be affected by the changes. In general, it is expected that the time since emission is relatively short in these transient plumes such that larger urban-scale to regional effects on EFs will have no impact.

**Page 14; Lines 3-5: Was the HNCO measurement technique identical in all these studies? Please clarify.**

The HNCO measurement technique was not the same for all studies (the technique used is listed in Table 3). Relevant to lines 3-5, Brady et al., (2014) used an Acetate-TOF-CIMS while Suarez-Bertoa and Astorga (2016) used FTIR. A line has been added to the text to explicitly state this:

 *"HCNO was measured by Acetate-TOF-CIMS and Fourier Transform Infrared Spectroscopy (FTIR) in the former (Brady et al., 2014) and latter (Suarze-Bertoa and Astorga, 2016) studies, respectively."*

**Conclusions: This section is very well written and some of the quantitative findings reported here can be put in the abstract which currently has scope for improvement.**

Thank you. A careful read of both the abstract and the conclusions indicates to us that all quantitative values are already contained in both sections.

**Figure 2: BKG instead of BCK in the Figure caption?**

Thank you, this has been fixed.

**Supplement: Page 2: The molecular formula of tricholorobenzene is incorrect. Should be C6 instead of C3 in the molecular formula.**

This was corrected.

**Also given the low mixing rations observed after background correction, the authors should explain the magnitude and methodology for the sensitivity corrections owing to changing humidity of sample air in more detail.**

The method for quantifying the humidity correction for the sensitivity for HCNO and HCN has been described in the SI. We have now included some additional detail in SI:

"*For these HR-TOF-CIMS measurements the humidity entering the ion-molecule region (IMR) is proportional to the m/z fragment ratios associated with iodide ($I^-$; m/z 127) and iodide clustered with water ($I \cdot H_2O^-$; m/z 145). Hence the ratio of m/z 127: m/z 145 during humidity calibrations is applied to the ambient data based upon the ambient m/z 127: m/z 145 ratio. The magnitude of the humidity corrections based upon the above ratio during the study ranged from 10-20% for both HNCO and HCN.*"

**Wren et al. present emissions findings from a short mobile laboratory deployment in the Greater Toronto Area (GTA) in the summer and winter of 2015-2016. Their manuscript results mostly focus on trends in ambient concentrations and emissions factors of black carbon (BC), isocyanic acid (HNCO), and hydrogen cyanide (HCN). A key finding of this study is that emission factors of BC, HNCO, and HCN are lower than those reported in the literature suggesting that the mobile source fleet in GTA is relatively clean. They also report that mobile sources contribute substantially to the ambient concentrations of HNCO and HCN in urban environments and may be more important than biomass burning in those regions.**

**The methods used, the presentation of results, and the conclusions based on the results are robust. The manuscript is very well written and easy to follow; amongst the top 10 percentile of manuscripts I have reviewed. I have very few comments on the manuscript (see below) and strongly recommend publication in ACP. Despite being limited to GTA, the methodology and results from this manuscript will be very useful to the air quality and atmospheric chemistry community.**

**1. Page 5, lines 36-38: The inlet appears to be quite long given that HNCO and HCN – that tend to be sticky molecules – might suffer large losses. Were the losses through this length and this tubing material quantified for HNCO and HCN? What are the implications of tube losses on the study? Also, if the material can stick and be released as and when the equilibrium between the material on the tube and in the tube is perturbed, would the measured delay result in miscalculations?**

Neither of these compounds are "sticky" and in fact are some of the easier species to measure in this regard. The only possibility for line loses may come about from their dissolution in any liquid water collected in the tubing. Given that the entire inlet was heated (See methods), this is an unlikely scenario. Furthermore, since measured plumes of HNCO and HCN very nicely overlapped peaks of $CO_2$ (with no line loses at all, after accounting for line length delays), it is highly likely that broadening or delays caused by equilibria were unimportant.

**2. Page 5, line 37: Easy to calculate but what was the residence time in the sampling line?**

The residence time for the sampling line was calculated to be ~ 0.25 s (this has been added in the main text). Note the residence times for instruments connected downstream were longer. As noted in the Supplement (Section 1.5), a time-offset was applied to the pollutant dataset to correct for differences in residence time.

**3. Page 6, line 20: When mentioning the supplement, can you specify the correct section in the supplement?**

Added the Supplement Section.

**4. Page 9, line 9: Include findings about agricultural burning from Chandra and Sinha (2016).**

Added a sentence to include this reference (also in response to Reviewer 1):

*"However, Chandra and Sinha (2016) report annual HNCO mixing ratios of 0.94 ppbv for a suburban site in the Indo-Gangetic Plain that is strongly influenced by crop-residue fires; a much higher average*

*summertime HNCO concentration of 1.7 ± 0.06 ppbv was recently measured at the same site (Kumar et al., 2018)."*

**5. The seasonal differences described in Section 3.1.1 and visualized in Figure 2 may not be directly interpretable based on differences in the absolute concentrations in the two seasons. For example, for the same emissions and sources, wintertime concentrations for a species can be higher simply from shallower boundary layer heights. So higher wintertime concentrations may not reflect changes in emissions or sources. On the same note, for the same source, ambient temperatures may result in very different emissions, e.g., Suarez-Bertoa et al.(2016) found vehicular emissions of HNCO to be much higher at lower temperatures. Furthermore, for the same source and emissions, photochemistry could influence the background concentrations and contributions to the total. Would it have been better to compare seasonal differences after ratio-ing the species of interest against an inert tracer such as CO?**

Unfortunately CO data is not available.  The potential influence of a shallower boundary layer height in the wintertime has been included to the discussion as per Reviewer 1 suggestions (relevant paragraphs of Section 3.1.1) for benzene, HNCO, and HCN:

*"Increases in wintertime benzene may also be attributed to a shallower boundary layer height.";
"Shallower boundary layer heights would be expected to lead to enhanced wintertime concentrations if HNCO emissions/sources remain constant, and yet we observe lower concentrations of HNCO in the winter. Hence the boundary layer height is a potential issue that can reduce the apparent differences between summer and winter"; "As with HNCO, despite shallower boundary layer heights in the winter, the overall concentrations are observed to be lower in the winter."*

In the case of HNCO and HCN (higher mixing ratios in the summer), the shallower boundary layer height in the winter could be reducing the apparent difference between the summer and winter, which we have now acknowledged in the text. A sentence has been modified to acknowledge the Suarez-Bertoa and Astorga (2016) results:

*"Although lower temperatures are thought to enhance HNCO vehicle emissions (particularly cold-start emissions) (Suarez-Bertoa and Astorga, 2016), the similarity in the magnitude of the LOCAL component between seasons suggests that, overall, the primary on-road HNCO emissions remain relatively constant."*

**6. Isn't the statement starting on page 10, line 42 about LOCAL versus BKG HCN also true for HNCO?**

This is only true for HNCO in the summer.  The sentence has been amended to acknowledge this:

*"The bulk of the total measured HCN concentration is in the BKG component rather than the LOCAL component, especially in the summer, suggesting that in relative terms, on-road HCN sources may be less significant than other regional or global sources.  This is in contrast to benzene (dominant LOCAL component in both seasons) and HNCO (dominant LOCAL component only in the winter)."*

**7. Section 3.3: Am I understanding this right that the emission factors are calculated only using the LOCAL estimates? Also, if the LOCAL estimates include emissions from near-road non-mobile sources, the emission factors in this work would serve as an upper bound?**

Yes, the emission factors are calculated using the local estimates. In the case of the plume-based analysis, the calculation is constrained to a measured plume and therefore ideally captures the mobile source only. The time-based analysis is more likely to integrate over other near-road and non-mobile sources, and this was mentioned in the Supplement Section 3.3 Text has been added to the main manuscript Section 3.2 (Comparison of plume-based and time-based emission factor methodologies) to acknowledge this:

*"Since the LOCAL component used in the analysis may also include near-road or non-mobile sources, the EFs calculated using this method likely represent an upper bound."*

**8. It isn't clear to me how one would go about doing this but given the different pollutants measured and differences in their relative proportions in gasoline versus diesel vehicles., there must be a way to apportion the various pollutants measured into their contributions from gasoline and diesel. This would be an interesting exercise with huge value to air quality managers/regulators.**

We agree that this would be ideal. However, it is impossible for us to differentiate gasoline and diesel EFs in this work (individual plumes cannot be attributed to a gasoline or diesel vehicle). The mobile measurements would have to be done differently (such as making measurements on roads with restricted HDDV traffic). We made the assumption that our measurements capture the proportion of gasoline and diesel vehicles in the fleet and therefore the EFs can be scaled up using combined gasoline and diesel fuel consumption. We did attempt to stratify the EFs by road type (expecting highways to have more diesel vehicles vs smaller roads) but initial results were not definitive and so we did not pursue this further.

**9. Section 3.3.3: Why was HCN compared in Table 6 in mg/km while HNCO was compared in Table 5 in mg/kg-fuel?**

The majority of the literature HCN EFs were reported in mg/km and so we used those units for the comparison, thereby avoiding having to make assumptions in converting reported EFs into mg/kg-fuel. On the other hand, the majority of HNCO EFs were reported in mg/kg-fuel (and this is the unit we measure in). Table 2 lists the EFs for both HCN and HNCO in both units and has a footnote regarding the conversion to mg/km.

**10. Assuming that the BKG estimates are representative of ambient concentrations away from the roadway, would it be safe to say that the HNCO concentrations in GTA are significantly lower than the 1 ppbv health threshold suggested by Roberts et al. (2011).**

Yes I would agree with this statement. A sentence has been added to Section 3.1 (Overview of mobile pollutant measurements) to acknowledge this:

*"Overall, the magnitudes of the HNCO mixing ratios in both seasons (~45 pptv and ~26 pptv for the summer and winter, respectively) are much lower than the 1 ppbv harm threshold (Roberts et al., 2011; Wang et al., 2007)."*